# A Novel Role of *Medicago truncatula* KNAT3/4/5-like Class 2 KNOX Transcription Factors in Drought Stress Tolerance

**DOI:** 10.3390/ijms241612668

**Published:** 2023-08-11

**Authors:** Maria Adelaide Iannelli, Chiara Nicolodi, Immacolata Coraggio, Marco Fabriani, Elena Baldoni, Giovanna Frugis

**Affiliations:** 1National Research Council (CNR), Institute of Agricultural Biology and Biotechnology (IBBA), Rome Unit, Via Salaria Km. 29,300, Monterotondo Scalo, 00015 Roma, Italy; mariaadelaide.iannelli@cnr.it (M.A.I.); chiara.nicolodi@cnr.it (C.N.); immacolatacoraggio1958@gmail.com (I.C.); marco.fabriani@cnr.it (M.F.); 2National Research Council (CNR), Institute of Agricultural Biology and Biotechnology (IBBA), Via Alfonso Corti 12, 20133 Milan, Italy; elena.baldoni@cnr.it

**Keywords:** KNOX2 transcription factors, drought stress, RNA interference, *Medicago truncatula*, proline dehydrogenase, cluster analysis, gene co-expression networks

## Abstract

Class 2 KNOX homeobox transcription factors (KNOX2) play a role in promoting cell differentiation in several plant developmental processes. In *Arabidopsis*, they antagonize the meristematic KNOX1 function during leaf development through the modulation of phytohormones. In *Medicago truncatula*, three KNOX2 genes belonging to the *KNAT3/4/5-like* subclass (*Mt KNAT3/4/5-like* or *MtKNOX3-like*) redundantly works upstream of a cytokinin-signaling module to control the symbiotic root nodule formation. Their possible role in the response to abiotic stress is as-of-yet unknown. We produced transgenic *M. truncatula* lines, in which the expression of four *MtKNOX3-like* genes was knocked down by RNA interference. When tested for response to water withdrawal in the soil, RNAi lines displayed a lower tolerance to drought conditions compared to the control lines, measured as increased leaf water loss, accelerated leaf wilting time, and faster chlorophyll loss. Reanalysis of a transcriptomic *M. truncatula* drought stress experiment via cluster analysis and gene co-expression networks pointed to a possible role of MtKNOX3-like transcription factors in repressing a proline dehydrogenase gene (*MtPDH*), specifically at 4 days after water withdrawal. Proline measurement and gene expression analysis of transgenic RNAi plants compared to the controls confirmed the role of *KNOX3-like* genes in inhibiting proline degradation through the regulation of the *MtPDH* gene.

## 1. Introduction

Climate change typically results in prevalent drought stress conditions over vast areas at a global scale, threatening agricultural production world-wide. Drought stress affects the plant growth rate and development by affecting several physiological and biochemical functions, ultimately leading to reduced productivity [1,2]. Drought stress alters plant growth and development and severely affects photosynthesis, chlorophyll content, nutrient metabolism, ion uptake, and carbohydrates metabolism [1,3,4]. Plants have evolved several morphological, physiological, biochemical, and molecular mechanisms to overcome drought stress effects [1]. Since cellular water loss marks the first event of drought stress, the first plant response aims at increasing water uptake and maintaining cellular water content. This is achieved by enhancing root growth and accumulating organic and inorganic solutes in the cytosol, such as proline and sugars, to lower cellular osmotic potential, thereby maintaining cell turgor and improving water uptake from the drying soil [5,6]. In particular, proline acts as an excellent osmolyte, but also as a metal chelator, an antioxidant, and a signalling molecule [7,8]. Mechanisms enhancing stomatal closure, the reduction in shoot growth, and the acceleration of leaf senescence are also triggered to limit water loss [2,9]. The regulation of plant stress responses is inherently coordinated with other essential biological processes, especially growth-related pathways [2]. A balance between plant growth and survival to stress must be achieved [9], and hormones play a key role in this by integrating developmental signals with those related to stress response [10]. Drought response also involves genome-wide transcriptional reprogramming that leads to additional protective mechanisms, such as osmotic adjustment, detoxification, the mitigation of stress-induced damage, and the amplification or attenuation of stress signalling [11]. Stress-specific transcription pathways are connected to upstream signalling and developmental pathways via transcription factors (TFs). Several families of the TFs have been identified as important players in orchestrating the response to drought [11,12,13,14,15,16], but only few in the delicate balance between growth and survival. Amongst the hormones involved in the drought stress response, cytokinins (CKs) are good candidates in the trade-off between plant growth and stress resilience [17]. CKs mediate drought resilience by modifying root architecture and improving root fitness, influencing the photosynthetic machinery, enhancing antioxidant mechanisms, modulating the water balance, and directly affecting the expression of drought-responsive genes [18].

KNOTTED1-LIKE homeobox (KNOX) TFs control several aspects of plant development by modulating hormonal homeostasis and integrating the signalling of cytokinins, gibberellin, brassinosteroids, and auxin [19,20,21,22]. Due to their involvement in controlling multiple hormonal pathways during plant development, they may constitute novel good candidates in the balance between growth and stress response. KNOXs belong to the Three Amino acid Loop Extension (TALE) protein family of the TFs (KNOX and BLH) which are found in all eukaryote lineages [23] and are classified into two subfamilies, class 1 (KNOX1) and class 2 (KNOX2). KNOX1 and KNOX2 were proposed to have antagonistic functions in development, with KNOX1 promoting meristematic cell fate and KNOX2 promoting cell differentiation [24]. In *Arabidopsis*, there are four *KNOX2* genes, *KNAT3, KNAT4, KNAT5* that are closely related to each other, and *KNAT7* [21]. So far, there are no studies that directly associate members of KNOX2 TFs with stress response, although there are some indirect indications that this could be the case [25,26]. KNAT3, together with its TALE interactor BLH1, was shown to mediate ABA responses during seed germination and post-germinative growth [25]. Additionally, *KNAT3* and *KNAT4* were amongst the 2000 drought-responsive genes identified in *Arabidopsis thaliana* under progressive soil drought stress using whole-genome oligonucleotide microarrays [26].

We decided to investigate the possible role of KNOX2 TFs in the response to drought stress in the model legume *Medicago truncatula*. In the *M. truncatula* genome, there are five *KNOX2* genes, of which four more are related to the *Arabidopsis KNAT3, KNAT4, KNAT5* (*Mt KNAT3/4/5-like* genes, also known as *MtKNOX3-like* genes), and one more is related to *KNAT7* [27,28]. *M. truncatula* is a plant that occurs naturally in the arid and semi-arid environments of the Mediterranean Basin area, and which has also been developed into an annual legume forage crop in Australia. It can be considered as a drought-adapted species and an ideal experimental model to advance our understanding of the underlying molecular mechanisms of drought adaptation and tolerance in legumes [29]. Legumes is the second family of crops after cereals in terms of cultivated areas worldwide, represents a valuable source of proteins for food and feed, and is vital to sustainable agriculture due to its ability to fix atmospheric nitrogen by establishing symbiotic interactions with rhizobia [29]. As observed for other crops, legume production is threatened by the extreme environmental conditions, with water scarcity being the main constrain [30]. Legume species vary in tolerance/sensitivity to drought, but in all cases, the final yield is significantly decreased [30].

Characterization of drought stress response in *M. truncatula* have been addressed, although functional studies to identify the important regulators are still limited. Global reprogramming of transcription and metabolism during progressive drought and after rewatering have been assessed in *M. truncatula* [29]. In this study, a detailed overview of the transcriptome and metabolome changes associated with the progression of drought stress, linked to the physiological measurements of the plant water status, was achieved in shoots and roots [29]. These analyses confirmed a role of abscisic acid (ABA) as one of the early signal molecules that modulate physiological responses to drought [31]. Among the metabolites detected in drought-stressed plants, myo-inositol and proline had striking regulatory profiles indicating involvement in *M. truncatula*’s drought tolerance [29]. The genes encoding Δ1-Pyrroline-5-carboxylate synthase (P5CS), a key enzyme in proline synthesis, and those coding for proline degrading enzymes, such as proline dehydrogenase (PDH), were up- and down-regulated, respectively. A comparison of molecular and physiological responses to drought stress in the two widely used *M. truncatula* genotypes, Jemalong A17 and R108, highlighted important differences between the two genotypes, with Jamalong A17 showing a greater tolerance to drought stress [32]. 

To explore the possible role of *KNOX2* genes in the *M. truncatula* response to drought, we used a previously obtained RNA interference (RNAi) construct [28] to produce transgenic lines in which four *MtKNOX3-like* genes (*MtKNOX3, MtKNOX5*, *MtKNOX9*, and *MtKNOX10*) were simultaneously knocked down. The *MtKNOX3-like* genes were previously shown to redundantly work upstream of a CK-signalling module to control the symbiotic root nodule formation [28]. They are homologous to the *Arabidopsis KNAT3/4/5* TFs and their expression is found in both root and leaf tissues [27,28]. When tested for response to water withdrawal in the soil, RNAi lines displayed a lower tolerance to drought conditions compared to controls. Reanalysis of the transcriptomic data from a drought stress time course experiment [29] using cluster analysis and Gene Co-expression Network (GCN) pointed for the first time to a possible role of KNOX3-like TFs in repressing a proline dehydrogenase gene at early stages of stress. Proline measurement and gene expression analysis of transgenic RNAi plants compared to the controls confirmed the role of *KNOX3-like* genes in increasing proline content by inhibiting its degradation through the repression of a *PDH* gene.

## 2. Results

### 2.1. Generation and Selection of MtKNOX3-like RNAi Transgenic Lines

To investigate a possible role of *M. truncatula* KNOX3-like TFs in drought response, two previously obtained *Mt KNAT3/4/5 RNAi constructs* (*Mt KNAT3/4/5-like RNAi* and *Mt KNAT3/4/5-like hox RNAi*) [28] were used to transform leaves and petals of *M. truncatula* R108 [33]. After selection on kanamycin, nine T0 plantlets for the *Mt KNAT3/4/5-like RNAi* construct (Figure 1a) and five from construct *Mt KNAT3/4/5-like hox RNAi* were analyzed for the presence of the kanamycin resistance gene (*nptII*). All the generated T0 plants resulted positively and were grown to obtain T1 seeds. However, no seeds were obtained from T0 plants transformed with *Mt KNAT3/4/5-like hox RNAi*, as leaves from these plantlets accumulated anthocyanin, lost chlorophyll, and died at an early developmental stage before producing seeds (Figure 1b). Plants transgenic for the *Mt KNAT3/4/5-like RNAi* construct and those transformed with the empty vector reached maturity and set seeds. The T1 plants were genotyped for kanamycin resistance and those that resulted positive were grown and analyzed for the gene expression of *MtKNOX3*, the most abundant *KNAT3/4/5-like* transcript [28], using quantitative real-time PCR (Figure 1c). 

Amongst the T1 lines harboring the resistance gene, some plants displayed a reduction of more than 60% in the expression of *MtKNOX3*. Further gene expression analysis showed that the *Mt KNAT3/4/5-like RNAi* construct was effective in down-regulating the expression of the four tagged *Mt KNAT3/4/5-like* genes (an example is shown in Figure 1d). T1 plants (p.23, p78 and p.88) from three independent T0 transformants, showing the lowest expression of *MtKNOX3* (CI38, CI73, and CI76, respectively)*,* were selected for further analysis on the T2 seeds. 

### 2.2. Physiological Characterization of MtKNAT3/4/5-like RNAi Transgenic Lines

To have a preliminary evaluation on whether the knock down of the *MtKNOX3-like* genes in *M. truncatula* affects drought response, leaves from the T1 progeny of the CI38, CI73, and CI76 T0 lines were also tested for water loss. *Mt KNAT3/4/5-like RNAi* leaves lost water more quickly than the empty vector controls, and the differences were already detectable after 16 h, with some lines showing 50% water loss compared to the 30% of control leaves (Figure 2a). After 40 h, leaves from all transgenic RNAi plants had lost more than 60% of water, some even reaching 80% of water loss, compared to the control plants that displayed no more than 50% water loss. These data pointed to a decreased drought tolerance of the *MtKNAT3/4/5-like RNAi* lines. To physiologically characterize the response of RNAi plants to oxidative stress, we took advantage of paraquat/methyl viologen. We observed a lower reduction in electrolyte leakage (60%) from CTR leaves exposed for 24 h to methyl viologen (Figure 2b) compared to the leaves of the RNAi lines (85%), thus indicating that the RNAi lines were less tolerant to the herbicide.

We then analyzed total chlorophyll content in detached leaves during a dark-induced leaf senescence assay at different time points (0 d, 6 d, and 10 d) (Figure 2c, left panel). Both empty vector plants (CTR) and *MtKNAT3/4/5-like RNAi* (RNAi) leaves showed a significant reduction in the content of total chlorophyll over time, but transgenic RNAi leaves lost pigment faster than the control plants. RNAi plants had lower total chlorophyll content already at time 0. The percentage of leaves undergoing senescence was significantly higher in the RNAi leaves compared to the controls at early time points (Figure 2c, right panel). The survival strategy behind the dark-induced senescence confirmed the reduction in the RNAi plants’ tolerance to stress, since its regulation paths are closely related [34,35]. Altogether, the responses of RNAi transgenic lines to different abiotic stress treatments indicated that the knock down of the *MtKNOX3-like* genes contributed to the increase in oxidative stress sensitivity.

### 2.3. Silencing of Mt KNAT3/4/5-like Transcription Factors Decreases Tolerance to Drought Stress

To validate these initial physiological and biochemical observations, we tested the behavior of T2 RNAi plants compared to the control plants for drought stress in soil at the emergence of the seventh leaf. The progeny of the T1 plants containing the vector only (from now on “control” plants) or the *Mt KNAT3/4/5-like RNAi* construct (from now on “RNAi” plants) were sown in soil and genotyped for the *nptII* kanamycin resistance gene. Plants harboring the constructs were grown in soil for five weeks under the regular watering regime. Progressive drought was induced by water withdrawing for 18 days in order to mimic what plants experience in their natural environment. During the experiment, soil moisture and soil electric conductivity were monitored with a sensor (as schematized in Figure 3a). Mild and severe stress timing was defined based on Zhang et al. 2014 [29]. At day 12, corresponding to mild stress, the leaves of RNAi plants started to wilt, whereas those of the control maintained a good turgor (Figure 3b). At day 18, corresponding to severe stress, RNAi T2 lines displayed rolled and dried up leaves, whereas the control plants were still green and only moderately wilted (Figure 3c). All these findings pointed to an increased sensitivity to drought stress of the RNAi lines, thus to a role of the *MtKNOX3-like* genes in drought stress tolerance.

### 2.4. Meta-Analysis of a M. truncatula Drought Stress Transcriptomic Experiment Identifies MtKNOX3-like Genes as Rapidly Induced during Drought Stress

In order to identify the regulatory networks that involve *MtKNOX3-like* genes, we explored publicly available transcriptomic data. We took advantage of a high-quality drought stress kinetics carried out and described in Zhang et al. 2014 [29], where transcriptomic and metabolomic response was analyzed at 2, 3, 4, 7, and 14 days after water withdrawal in *M. truncatula* shoots and roots. The experiment used the Affymetrix Medicago Gene Chip from which we extract functional modules of interest (all TFs modules [36]; CK, ABA, and gibberellin (GA) hormonal modules; and oxidative stress and proline metabolism modules). In fact, for the analysis, we applied a targeted knowledge-based bioinformatic pipeline that we previously described and successfully used to partition the data and focus on specific functional genetic modules [14,16,37]. Overall, 2468 probes, as listed in Appendix A, were considered and used for the K-means clusterization of gene expression to explore the main regulatory transcriptional networks acting during induction of drought stress in the shoot. The clustering gene expression data allows to identify substructures in the data and identify groups of genes that may share regulatory relationships. Co-expression and anticorrelation are commonly used in bioinformatics pipelines to identify potential TF targets [38,39,40,41], as target genes are often co-expressed with the TFs that positively regulate them, or are anticorrelated with the TFs acting as negative regulators.

The *MtKNOX3-like* genes were represented by the following Affy probes: Mtr.37567.1.S1_at (*MtKNOX3*); Mtr.6737.1.S1_s_at (*MtKNOX5*); Mtr.8750.1.S1_at (*MtKNOX10*); and Mtr.8842.1.S1_at (*MtKNOX9*). We identified ten main gene expression clusters that are shown in Appendix A (K1–K10). Three out of the four *MtKNOX3-like* genes (*MtKNOX3, MtKNOX5,* and *MtKNOX10*) were placed in a specific cluster of 180 genes that are upregulated at 4 d of drought stress only (Cluster 3, Appendix A). This cluster was highly anticorrelated (r = −0.84) with a cluster of 175 genes (Cluster 6, Appendix A) that were specifically downregulated at 4 d of drought stress. In Clusters 3 and 6, 58 and 61 genes, respectively, had a cluster score ≥0.8 that identifies those genes whose expression closely match the core and were considered for further analysis (Appendix A). Most of the genes placed in two pairs of the anticorrelated clusters that contained the genes either gradually increased (Cluster 1 and Cluster 9) or gradually decreased (Cluster 7 and Cluster 10) their expression during drought stress (Appendix A). *MtKNOX9* was placed in Cluster 9, which contains the genes that are initially induced after 4 days of water withdrawal and steadily increase during stress. 

We focused on Clusters 3 and 6 to explore the Gene Co-expression Networks (GCN) underlying the early expression changes occurring at day 4 in drought-stressed shoots that may involve *MtKNOX3-like* genes (Figure 4a). To do this, we used the transcript expression values from the 27-microarray data-related shoot samples to perform a robust pairwise correlation analysis and to identify those genes that resulted highly co-expressed or highly anticorrelated with the *MtKNOX3-like* genes (the blue and orange lines are shown in Figure 4b, respectively). Since KNOX TFs work as heterodimers with BLH TFs, we were also interested to explore KNOX and BLH genetic interactions during drought response. Hence, we selected all the transcripts representing both KNOX and BELL/BLH (superfamily of TALE TFs) for the construction and analysis of the GCN [42]. A total of 145 genes (Appendix A) were included to perform pairwise correlation analysis of gene expression values (Appendix A). The genes with significant correlations (*p* value ≤ 0.05) and absolute Pearson’s correlation (|r|) values ≥ 0.6 were used to develop the GCN. The final network included 140 genes and had a high clustering coefficient (0.619) and a characteristic path length of 1.977 (Appendix A).

To determine the relationships among the selected genes and to identify the hub genes in the early response to drought stress at 4 d in the shoot (Figure 4a), the topological features of the network were analyzed (Appendix A). We first looked at the genes displaying the highest node degree, which is the number of neighbors to which a node directly connects and characterizes the hub genes in a network [43]. The top five probes in the list belonged to Cluster 6 and represented two genes: four probes (Mtr.12290.1.S1_at, Mtr.12290.1.S1_s_at, Mtr.12291.1.S1_s_at, and Mtr.42984.1.S1_at), with degrees of 77, 78, 78, and 79 respectively, all represented one proline dehydrogenase 1 (namely *MtPDH_1,2,3,4* in the GCN) gene (Mt v4 ID: *Medtr7g020820;* Mt v5 ID: *MtrunA17_Chr7g0221391*), encoding a key enzyme for proline degradation, of which its suppression induces proline accumulation [29]; one probe (Mtr.41252.1.S1_at) represented a gene predicted to encode for a BLH TF (Mt v4 ID: *Medtr1g023050*; Mt v5 ID: *MtrunA17_Chr1g0155681*) (degree = 76) that we named *MtBLH6_2* in the GCN. *MtKNOX3* and *MtKNOX5* also showed a high node degree (51 and 59, respectively). They resulted highly co-expressed with four BELL/BLH genes (namely *MtBLH1_BLH5*, *MtBLH1*, *MtBLH6_2,* and *MtBELL1_2* in the GCN), and anticorrelated with the *PDH* gene. *MtKNOX3* and *MtKNOX5* shared most of the connections (44). The subnetwork formed by the *MtKNOX3* and *MtKNOX5* genes stayed at the core of the GCN and is shown in green in Figure 4a. Probes, gene IDs, and the relative Pearson’s correlation of the genes shared by *MtKNOX3* and *MtKNOX5* are shown in Appendix A. The connections shared by all the three *MtKNOX3-like* genes in Cluster 3, *MtKNOX3, MtKNOX5,* and *MtKNOX10*, were 26 and included the negative correlation with the *MtPDH* gene. KNOX2 were reported to act as transcriptional repressors [24], therefore possible *MtKNOX3-like* target genes would likely place in the anticorrelated Cluster 6. In this regard, the *MtPDH* gene, whose probes are highly anticorrelated with *MtKNOX3, MtKNOX5*, and *MtKNOX10*, may constitute a valuable downstream target that would link the *MtKNOX3-like* genes to drought response. The expression of the most abundant *MtKNOX3-like* and *BLH* genes identified in Cluster 3 were also analyzed in the transcriptomic data of Zhang et al. [29] in the roots sampled at the same time points of the shoot analysis in comparison to the *MtPDH* gene (Figure 5). The *KNOX/BLH* genes had a similar expression in the roots, with a transient increase at day 4 of drought stress. This increase corresponded to a decrease in the expression of the *MtPDH* gene, as observed in the shoot. Therefore, the opposite expression pathways of the *MtKNOX3-like* and *MtPDH* genes were conserved in both the shoot and roots of the plants under drought stress.

### 2.5. Gene Expression Analysis to Validate Clustering and GCN Analysis Predictions

The *MtPDH* gene acts as a main hub in the GCN constructed with Clusters 3 and 6 that represents the genes whose expression either increases or decreases specifically after 4 days of drought stress. To assess whether MtKNOX3-like TFs may directly or indirectly regulate the expression of *MtPHD*, we analyzed the expression of the *MtPDH* transcript in the leaves of the control and RNAi T2 plants sampled at early stages in our water withdrawing experiment. At time 0, RNAi lines already displayed a 1.4-fold significant increase in *MtPDH* expression with respect to the control lines. At 4 d of drought stress, *MtPDH* transcript abundance decreased in the control plants, but was instead still higher in RNAi plants (an example of a representative RNAi line is shown in Figure 6). This difference was only observed at 4 days because at 8 days of drought, RNAi plants also displayed a reduction in *MtPDH* expression with respect to the watered plants, similarly to the control plants. This confirmed previous transcriptomic data from Zhang et al. [29], as well as confirmed the predicted role of *MtKNOX3-like* genes in contributing to rapidly repress *MtPDH* gene expression at very early stages of drought stress response.

### 2.6. Mt KNAT3/4/5 like-RNAi Plants Accumulate Less Proline during Drought Stress

At the transcriptional level, the knock down of *MtKNOX3-like* genes seems to delay downregulation of the proline dehydrogenase gene *MtPDH* at early stages of drought stress. To assess whether RNAi lines were affected in proline accumulation upon water withdrawal, proline content was measured and compared in the leaves of either the control or *Mt KNAT3/4/5-like RNAi* lines at 4, 8, and 12 days of drought stress (Figure 7).

Since the beginning of the drought stress, a slight increase in proline was observed after 4 and 8 days in both CTR and RNAi plants. From 8 d to 12 d of drought stress, a significant proline increase was observed. The proline levels in CTR plants showed a 7-fold increase, whereas the increase in RNAi plants was only of 3.5-fold. The overall reduction in proline accumulation observed in the RNAi plants may relate to a delay in downregulating the *MtPDH* gene at early stages of drought stress due to the *MtKNOX3-like* genes’ knock down and may account for the reduced tolerance to drought stress.

### 2.7. KNOX/BLH Binding Site Motifs in the MtPDH Regulatory Sequences

We searched the *MtPDH* gene, 3000 bp upstream nucleotide sequences, and intronic regions for possible KNOX/BLH binding sites. Since no binding site specificity information is available for KNOX2 TFs of the KNAT3/4/5 clade, we used the binding sites’ degenerated sequences that were previously described for KNOX1 and BLH proteins [44]. We searched for three regulatory motifs that we named KNOX/BLH_bs1 (KGACM), KNOX/BLH_bs2 (TGAYTGA), and KNOX/BLH_bs3 (TGATKKGA) (Appendix A). The results are shown in Figure 8 and Appendix A. In total, 23 KNOX/BLH binding motifs were found in the *MtPDH* gene, of which 14 in the promoter and 9 in the intronic regions. In particular, the third intron (187 bp length) of the *MtPDH* gene was strikingly enriched in KNOX/BLH binding sites (six binding sites) and may constitute a main regulatory element that could be tested in further analysis to validate MtKNOX3-like TFs and the *MtPDH* regulatory relationship.

## 3. Discussion

We investigated the possible role of MtKNOX3-like TFs that are members of the TALE superfamily of homeobox proteins in drought stress response. *MtKNOX3-like* genes belong to the *KNAT3/4/5-like* clade of KNOX2, which are highly co-expressed during plant development in both *Arabidopsis* and *M. truncatula,* thus pointing to functional redundancy. *Arabidopsis* single mutants for *KNAT3/4/5* genes do not display any phenotype [24]. In our previous work on the role of *KNAT3/4/5-like* genes in *M. truncatula* nodule symbiosis, we observed no phenotype when only *MtKNOX3* was knocked down [28]. Moreover, we observed that downregulating *MtKNOX3* would result in an overexpression of *MtKNOX5* and an overexpression of *MtKNOX3* associated with a reduced expression of *MtKNOX9* and *MtKNOX10*, thus suggesting strong transcriptional feedback regulation amongst the *KNAT3/4/5-like* genes [28]. Therefore, we used RNAi lines in which four *MtKNOX3-like* genes have been knocked down. One of the RNAi constructs we used did not allow the recovery of viable plants, whereas a second construct allowed us to obtain plants producing seeds that were less tolerant to drought stress. Biochemical and physiological parameters, analyzed under well-watered or drought conditions, pointed to a reduced tolerance of RNAi plants to stress compared to the control plants.

A role for *MtKNAT3/4/5-like* genes in root symbiotic nodule formation was recently shown [28]. Being recognized as developmental regulators, their possible role in the response to abiotic stress was neglected and poorly investigated so far. One study in *A. thaliana* linked KNAT3 and BLH1 TFs to ABA response. *BLH1* was induced by ABA and the encoded protein physically interacted with KNAT3 to modulate ABA sensitivity during seed germination and early seedling development [25]. *Arabidopsis KNAT3* and *BLH1* were also found amongst the genes upregulated during drought stress response in a microarray experiment [26]. Whereas no further study reported possible roles of the AtKNAT3/4/5 subfamily in stress response, several BLH TFs were instead related to different type of stresses in cotton, rice, and soybean [46].

We re-analyzed the transcriptomic data of drought stress in *M. truncatula* with a novel targeted knowledge-based approach using clustering and GCN analysis to predict possible targets of MtKNOX3-like TFs. The clustering gene expression data allows to identify the groups of genes that may act in the same pathway/response and may share gene function.

Our targeted cluster analysis identified 10 clusters characterized by different expression patterns. It has been observed that the genes whose expression patterns are strongly anticorrelated can also be functionally related [47]. This is the case for transcriptional repressors and their target genes that may cluster into separate anticorrelated groups. Since KNOX2 TFs are predicted to be transcriptional repressors, we focused on the cluster where most of the a*MtKNOX3-like* genes are placed (Cluster 3), and on the anticorrelated one (Cluster 6). The genes in Clusters 3 and 6 were characterized by a transient increase or decrease in expression, respectively, at 4 d of drought stress. Three out of the four *MtKNOX3-like* genes targeted by our RNAi constructs, i.e., *MtKNOX3, MtKNOX5*, and *MtKNOX10*, were placed in Cluster 3 and resulted highly anticorrelated to a *PDH* gene (*MtPDH*) placed in Cluster 6. This negative regulatory relationship between the three *MtKNOX3-like* genes and *MtPDH* was confirmed in our transgenic RNAi lines, which displayed a constitutive higher expression of *MtPDH* in leaves and a delay in *MtPDH* downregulation at early stages of water deprivation with respect to control plants.

It was clearly shown that *M. truncatula* tightly regulates proline production and accumulation during drought stress progression by the upregulation of several genes encoding P5CS, a key enzyme in proline synthesis, and concomitantly repressing genes coding for proline degrading enzymes such as PDH [29]. *P5CS* genes were progressively induced during drought stress, whereas *PDH* genes were specifically downregulated at day 4. In our analysis, the expression pattern of *P5CS* genes corresponded to Cluster 1 (Appendix A), whereas the *PDH* genes were placed in Cluster 6. Zhang et al. [29] also showed that proline metabolite steadily accumulated in both shoots and roots from 4 d onwards. However, root proline content under drought stress was several folds higher than in shoots, despite the higher expression levels detected for *P5CS* in the shoots. All these findings suggested that proline accumulation in different organs and tissues may depend on the equilibrium amongst catabolism, biosynthesis, and transport. Proline metabolism is strongly connected to many cellular processes and is considered a regulatory hub of signaling pathways [48]. For this reason, a complex regulation of the expression of the genes involved in proline metabolism is expected.

Our cluster analysis successfully identified *MtPDH* as an important hub of the regulatory network that acts specifically, and transiently, at 4 days after water depletion. This regulatory network involves gene encoding of three MtKNOX3-like TFs and their several possible BELL/BLH interactors as potential repressors of *PDH* genes and may represent an important early switch in the response to water scarcity. We identified several KNOX/BLH binding sites in the regulatory regions of the *MtPDH* gene. The KNOX/BLH binding sites were found in the 3000 bp upstream of the *PDH* gene and intronic regions. Strikingly, the third intron of the gene was enriched in KNOX/BLH motifs. These findings suggest that the regulatory relationship between MtKNOX3-like TFs and *MtPDH* may be direct, although further analysis would be needed to confirm this hypothesis. No direct negative regulators of *PDH* have been identified so far. Several bZIP TFs are known to induce *PDH* expression [48], whereas only one mechanism to negatively regulate *PDH* expression has been demonstrated [49]. Veerabagu et al. showed that the *Arabidopsis* type-B response regulator 18 (ARR18) physically interacts with bZIP63 and negatively interferes with the positive transcriptional activity of bZIP63 on the *PDH1* promoter.

As mentioned above, two proline biosynthetic genes (P5CS) (Mt v4 ID: *Medtr4g020110*, Mt v5 ID: *MtrunA17_Chr4g0008951*—Mt v4 ID: *Medtr7g063650*; Mt v5 ID: *MtrunA17_Chr4g0008951*), represented in the microarray by three probes (Mtr.33511.1.S1_at, Mtr.42902.1.S1_s_at, and Mtr.11510.1.S1_at, respectively) were placed instead in Cluster 1 (Appendix A). This cluster is characterized by the genes that increased as drought stress progressed, with a peak expression at severe drought stress. They all had a high cluster score (> 0.9) and may also represent important hubs in the regulatory network formed by the anticorrelated Cluster 1 and Cluster 7 (Appendix A). However, these two clusters are not related to most *MtKNOX3-like* genes and their BELL/BLH partners, and there are no indications that they could be part of the KNOX2/BLH regulatory pathway.

In plants, the variation in proline anabolism and catabolism gene expression directly relates to the high proline accumulation [48,50,51]. No proline accumulation is detected under well-watered conditions, thus implying that proline accumulation is specifically induced by low water potential. Increasing the abundance of specialized metabolites and associated primary metabolites, such as ascorbates and proline, are considered as a marker for enhanced biotic and abiotic stress tolerance [8]. Indeed, transgenic RNAi lines displayed a delay in the accumulation of proline with respect to the controls, confirming a possible role of MtKNOX3-like TFs in rapidly switching off the proline degradation genes to speed proline accumulation in response to stress and to protect the cell membranes from drought-induced osmotic stress.

During drought stress, both osmotic and oxidative stresses are induced [7]. Proline and sugars act as osmolytes and scavengers of reactive oxygen species (ROS). They keep osmotic balance and cell turgor, stabilize membranes, and consequently prevent electrolyte leakage. They also bring ROS concentrations to normal ranges, thereby preventing oxidative bursts in plants [7,8].

*MtKNOX3-like RNAi* lines, besides being less tolerant to drought, rapidly underwent dark-induced senescence and were less tolerant to oxidative stress when subjected to oxidative paraquat treatment. These findings indicate that *MtKNOX3-like* genes may have a role in responding to other stress, involving oxidative bursts, through the modification of proline catabolism.

Overall, our approach, that integrated in planta functional studies and molecular and physiological phenotyping with a targeted knowledge-based bioinformatic analysis, was successful in identifying a novel KNOX/PDH regulatory module acting at early stages of drought stress perception. In our hands, partitioning the complex transcriptomics data based on deep knowledge of gene function and relative pathways allows to analyze subsets of data and reduce noise in the bioinformatic analysis, thus prompting to explore novel genetic relationships in complex biological networks.

The novel role in drought tolerance identified for KNOX2 TFs, until now known as players in developmental processes, opens new intriguing questions about genetic networks acting in the trade-off between abiotic stress response and growth.

## 4. Materials and Methods

### 4.1. Plant Material, Growth Conditions, and Stress Treatment

*Medicago truncatula* genotype R108 seeds were provided by INRA Montpellier, SGAP Laboratory, Mauguio, France. Seeds were scarified with sandpaper, sterilized for 15 min in bleach (12% [*v*/*v*] sodium hypochloride, Dexal, Eurospin IT), rinsed 5–6 times in sterile water, and vernalized in the dark at 4 °C in water-containing Petri dishes. After 2 days, the seedlings were transferred to 20 °C in the dark for germination. After 24 h, the germinated seedlings were transferred to soil (Cactaceae soil S.Q. 12, Vigor Plant Italia) and grown in a pot at 23–26 °C under long-days conditions (16 h light/8 h dark at 100 μmol/m^2^/s) and 40% relative humidity. For standard growing conditions, the plants were watered from the soil top every third day by alternating between water and a nutritive solution (3 mL/L Bayfolan Universale N/P/K, Bayer garden).

*M. truncatula* germinating seeds were selected and sown in squared plastic pots (9 × 9 × 10 cm with 0.6 L in volume), filled with pre-autoclaved soil (Cactaceae soil S.Q. 12, Vigor Plant Italia), and grown at 23–26 °C under long-days conditions (16 h light/8 h dark). The photon flux density at soil level was 100 μmol/m^2^/s and supplied mainly with cool light. The plants were watered from the soil top every second day, alternating between water and a nutritive solution (3 mL/L Bayfolan Universale N/P/K, Bayer garden) until 24 days after planting. At that time, most of the plants showed an emergence of the seventh leaf and an expanded sixth leaf. Half of the plants were put under drought-stress imposition by withholding watering, while the remaining half were kept with regular watering. For RNA isolation and metabolite/proline/sugar analysis, pools of the third, fourth and fifth fully expanded leaves were sampled in biological triplicates for both the control and drought-stressed plants at all time points. The leaf samples were frozen in liquid nitrogen and stored at −80 °C prior to RNA isolation and metabolite analysis. Soil humidity was measured daily via the Flower Care Smart Sensor (http://www.huahuacaocao.com/product (accessed on 29 June 2023)) and soil water loss was assessed via daily weighting watered and drought-stressed reference plant pots.

### 4.2. Plasmid Construction and Plant Transformation

Constructs for RNA interference (RNAi) of RNAi constructs targeting multiple *M. truncatula KNAT3/4/5-like* genes were previously described [28]. For stable plant transformation, two RNAi constructs targeting multiple KNOX2 genes, one based on the KNOXII domain of the MEINOX (131 bp) (*Mt KNAT3/4/5-like RNAi*) and one as part of the homeodomain (HOX) (134 bp) (*Mt KNAT3/4/5-like hox RNAi*) [28], were introduced into the *Agrobacterium tumefaciens* strain EHA105 [52] via electroporation.

*M. truncatula* Agrobacterium-mediated genetic transformation and plant regeneration were done according to Ratet and Trinh (EMBO practical course on the New Plant model System Medicago truncatula. 19 November–1 December 2001, Gif-sur-Yvette, France) [33] using healthy, young expanded leaves from 4 to 6-week-old plants, with some modifications. An additional step was introduced to minimize Agrobacterium overgrowth after the 2 days of leaf explant cocultivation. The plant material was washed 4–5 times with sterile water containing 400 mg/L of Augmentin, blotted dry on sterile filter paper and transferred to a SH3a solid medium. This simple additional step prevented Agrobacterium overgrowth and a good number of viable calli were obtained, most of which gave rise to somatic embryos within 3 weeks. Emerging embryos were transferred to a fresh medium containing 40 mg/L of kanamycin for selection.

### 4.3. Plant Genotyping and Selection of Transgenic Plants

*M. truncatula*’s positive primary transformants (T0) and their T1 and T2 progeny harboring the RNAi constructs were identified via a direct PCR amplification from cotyledon or through young leaf tissues using the Phire Plant Direct PCR Kit (Thermo Scientific, Waltham, MA, USA). A small piece of plant material was cut using a sterile pipette tip, placed in 20 μL of dilution buffer and crushed by pressing it briefly against the Eppendorf tube wall. For DNA amplification, 0.5 uL of the supernatant was used as template for a 20 μL PCR reaction containing Phire Plant PCR Buffer (including dNTPs and MgCl_2_), 0.5 μM of each primer, and 0.4 μL of the Phire Hot Start II DNA Polymerase. Amplifications were carried out in a thermal cycler (Eppendorf Mastercycler Personal, Hamburg, Germania) with cycling conditions as follows: 98 °C for 5 min; 40 cycles at 95 °C for 5 s, 55–65 °C for 5 s, and 72 °C for 20 s; and final extension at 72 °C for 1 min. The primers used are shown in Appendix A.

### 4.4. Physiological and Biochemical Measurements

Water loss was measured as previously described [53]. Ten fresh trifoliate expanded leaves from well-watered 5-week-old *M. truncatula* plants were detached from three T1 plants for each T0 original transformed plant and immediately weighed. The leaves were then placed in ventilated petri dishes and incubated in the growth chamber at 23 °C under continuous light. Fresh weights of the leaves were measured at time 0 and at designated time intervals (16, 20, 24, and 40 h). Water loss was calculated on the basis of the initial weight of the plants. All plant materials were sampled at 9:00, 3 h after the beginning of the day cycle.

To evaluate the oxidative stress tolerance, the excised leaf discs obtained from CTR and RNAi *M. truncatula* 5-week-old plants were subjected to methyl viologen (paraquat). Paraquat is the common name of the herbicide methyl viologen (MV; N,-N′-dimethyl-4,-4′-bipyridinium dichloride), which acts in the production of reactive oxygen species (ROS) via a light dependent mechanism. The ion leakage out of the leaf discs, due to the destruction of membrane lipids, was measured as an increase in the conductance of the floating solution, as described by Kasajima et al. (2017) [54].

For inducing the senescence of the detached leaves in the darkness, two to three leaves from 5-week-old *M. truncatula* plants were excised and placed in Petri dishes, wrapped with double-layer aluminum foil, and then kept at 22 °C. Biological triplicates were used. The total chlorophyll content was measured in the detached leaves at the initial experimental stage (0 d) and after 6 and 10 days, as previously described [37].

Proline accumulation was measured on pools of the third, fourth, and fifth fully expanded leaves, sampled in biological triplicates for both the control and drought-stressed plants at 0, 4, 8, and 12 days, following Lee et al.’s (2018) procedure [55].

### 4.5. Quantitative RT-PCR Analysis of Gene Expression

For quantitative real-time PCR (qRT-PCR) gene expression analysis, sample homogenization was performed with a TissueLyser (Qiagen, Hielden, Germany) following the manufacturer’s procedures. Total RNAs were extracted from the frozen leaves of *M. truncatula* using the Trizol reagent (Sigma-Aldrich, Darmstadt, Germania, http://www.sigmaaldrich.com/ (accessed on 29 June 2023)) and treated with DNase I (Qiagen, Hielden, Germany). First-strand cDNA was synthesized from 1.5 μg of the total RNA using the Superscript III first-strand synthesis system (Invitrogen, Darmstadt, Germania). Primer design was performed using Primer3 software (available at http://frodo.wi.mit.edu/cgi-bin/primer3/primer3_www.cgi (accessed on 29 June 2023)). qRT-PCR reactions were performed using the Eco Real-Time PCR System (Illumina, San Diego, CA, USA) following the manufacturer’s instructions, using 50–70 ng of the template cDNA (Biotool easy mix with an Eva green fluorophore) and 300 nM of the final primer concentration. Cycling conditions were as follows: 95 °C for 10 min; and 40 cycles at 95 °C for 15 s, 60 °C for 15 s, and 72 °C for 15 s. Three technical replicates and two independent biological experiments were performed in all cases. A ratio with *MtEF1-α* reference genes was used for the different experiments (the primers used are listed in Appendix A). The ratio value of the experimental control condition was set up to 1 as a reference to determine the relative expression of the fold change.

### 4.6. Gene Expression Meta-Analyses and Microarray Pathway Annotation

The transcriptomic data of either shoot or root drought response series from the article of Zhang et al. [11], based on the Affymetrix Medicago GeneChip containing 50,900 probes, were retrieved from the *M. truncatula* Gene Expression Atlas (http://mtgea.noble.org/v2/ (accessed on 29 June 2023)). The genes encoding the TFs were annotated and classified by family based on the TF annotations in the PlantTFDB database [36]. Gene IDs were retrieved by the Expression Atlas Server (https://lipm-browsers.toulouse.inra.fr/pub/expressionAtlas/app/mtgeav3/ (accessed on 29 June 2023)) [56,57]. Specific pathways were annotated based on the key words analysis, current literature, and homology with the *A. thaliana* genes. Different modules were annotated: TF, CK, GA, ABA, oxidative stress, and proline metabolism. The analyses started from the microarray values of 2468 selected probes representing the pathways of interest, from which the expression data were extracted and used to create a starting sub-database (Appendix A).

### 4.7. Correlation Analysis and Gene Co-Expression Networks

The pipeline for the k-means cluster analysis was performed according to Testone et al. (2019) [37] with some modifications. Briefly, mean values of the expression data for the 2468 selected probes from the drought stress shoot samples [29] were log-transformed using log2(x + 1) for data normalization. The optimum number of clusters was determined based on the converging results of the sum of squared errors (SSE) estimate and the Calinsky criterion [37]. Data scaling, k-means clustering, and visualization were performed in R as previously described [37]. The cluster analysis data are available in Appendix A. Clusters 3 and 6 were selected for the GCN’s construction, and the expression data for the TALE genes were also added. From the selected clusters, only the genes with a cluster score ≥ 0.8 were used to perform pairwise correlation analysis of the expression values from the biological triplicates. The expression data were log-transformed using log2(x + 1) for normalization, and Pearson pairwise correlation analysis was conducted across the selected samples using the “corrplot” and “hclust” packages of R software (https://rdrr.io/cran/corrplot/man/corrplot-package.html (accessed on 29 June 2023)). Significant correlations (*p*-value ≤ 0.05) with a Pearson’s correlation coefficient (r) ≥ |0.6| were used to develop the GCNs. The Cytoscape software platform v. 3.5.1 [42] (Shannon et al., 2003) was used to visualize the networks, to determine the relationships among the selected genes, and to identify the hub genes in the orchestration of the response to water withdrawing at day 4.

### 4.8. KNOX/BLH Binding Sites Analysis

To identify the putative cis-acting elements of KNOX/BLH TFs in *MtPDH* gene sequences, the 3000 bp length located upstream of the gene and the intronic regions were retrieved from the genome of *M. truncatula* (Mt4.0 v1) using Phytozome 13 [58] (Appendix A). Promoter and intronic sequences were then searched for specific motifs using RSAT Plants [59]. The motifs searched were the following: KNOX/BLH_bs1 (KGACM); KNOX/BLH_bs2 (TGAYTGA); and KNOX/BLH_bs3 (TGATKKGA).

## Figures and Tables

**Figure 1 ijms-24-12668-f001:**
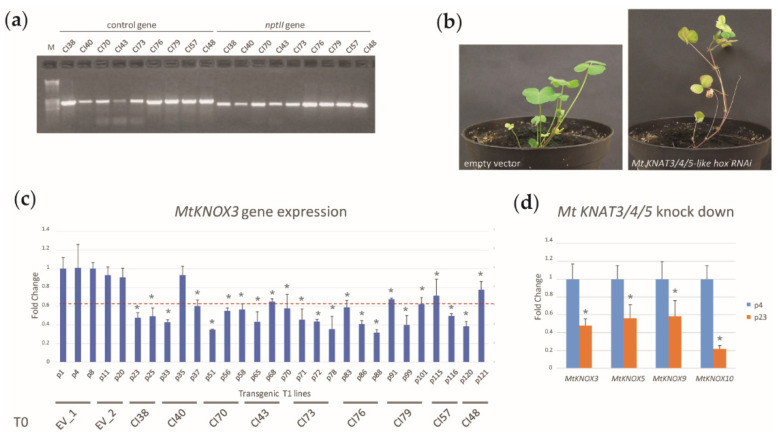
Selection and molecular analysis of *Mt KNAT3/4/5-like RNAi* transgenic lines. (**a**) Gel electrophoresis of PCR-amplification of DNA from transgenic T0 lines (CI) with either a control gene (*MtKNOX6*) or the *nptII* gene for the kanamycin resistance. Lane 1 = 1 kb DNA ladder; lane 2–10 = control *MtKNOX6* gene; and lane 11–19 = *nptII* kanamycin resistant gene. (**b**) Phenotype of plants transformed with the *Mt KNAT3/4/5-like hox RNAi* construct. (**c**) Quantitative real-time PCR analysis of *MtKNOX3* in the leaves of the T1 progeny of transgenic lines transformed with the empty vector only (EV1-2) or with the *Mt KNAT3/4/5-like RNAi* construct. The dotted red line indicates a 60% reduction in gene expression with respect to the empty vector plants. (**d**) Quantitative real-time PCR analysis of *MtKNOX3-like* genes in the leaves of the T1 progeny of transgenic lines transformed with the empty vector (p4) or with the *Mt KNAT3/4/5-like RNAi* construct showing the occurrence of silencing in the four tagged *MtKNOX3-like* genes *MtKNOX3, MtKNOX5, MtKNOX9*, and *MtKNOX10* (p23 = T2 RNAi line). Representative lines are shown. The expression level of genes was calibrated relative to the expression level of the empty vector and expressed in a fold change. The data represents the average of two biological replicates with three technical replicates each. Error bars represent + SD and asterisks (*) represent significant differences relative to the control (Student’s *t*-test: *p* ≤ 0.01).

**Figure 2 ijms-24-12668-f002:**
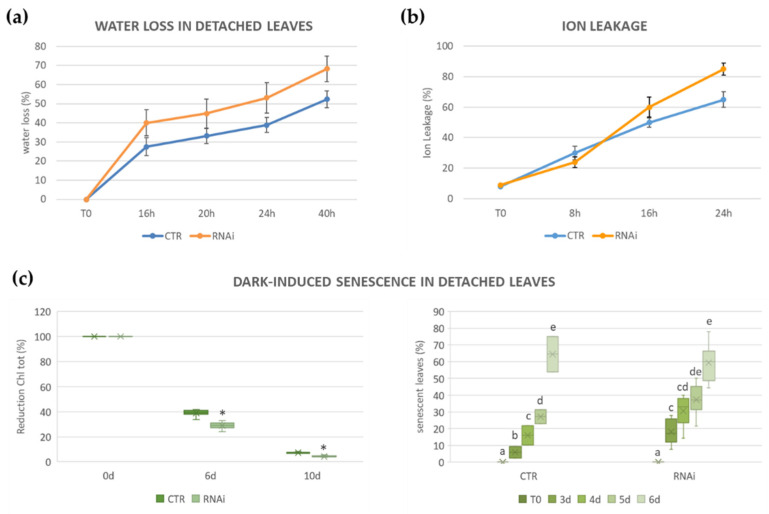
Physiological and biochemical responses of *MtKNOX3-like* RNAi transgenic plants. (**a**) Experiment of water loss in detached leaves of transgenic T1 lines (empty vector control, CI38, CI73, and CI76). *Y*-axis represents the percentage of the leaves’ water loss over time. Paraquat assay (**b**) and dark-induced senescence experiment (**c**) in detached leaves of transgenic T1 lines (RNAi and empty vector CTR). In (**b**), the *Y*-axis represents the percentage of the leaves’ ion leakage increasing over time. In (**c**), the graphs represent the percentage of the total chlorophyll decrease (box plot on the left), and the percentage of the leaves undergoing senescence at different times. The means of three independent RNAi lines from two plants for each biological replicate are shown. In the left panel, asterisks (*) indicate significant differences relative to the control (Student’s *t*-test: *p* ≤ 0.01). In the right panel, different letters indicate significant differences between the samples (Tukey’s test, *p* value ≤ 0.05).

**Figure 3 ijms-24-12668-f003:**
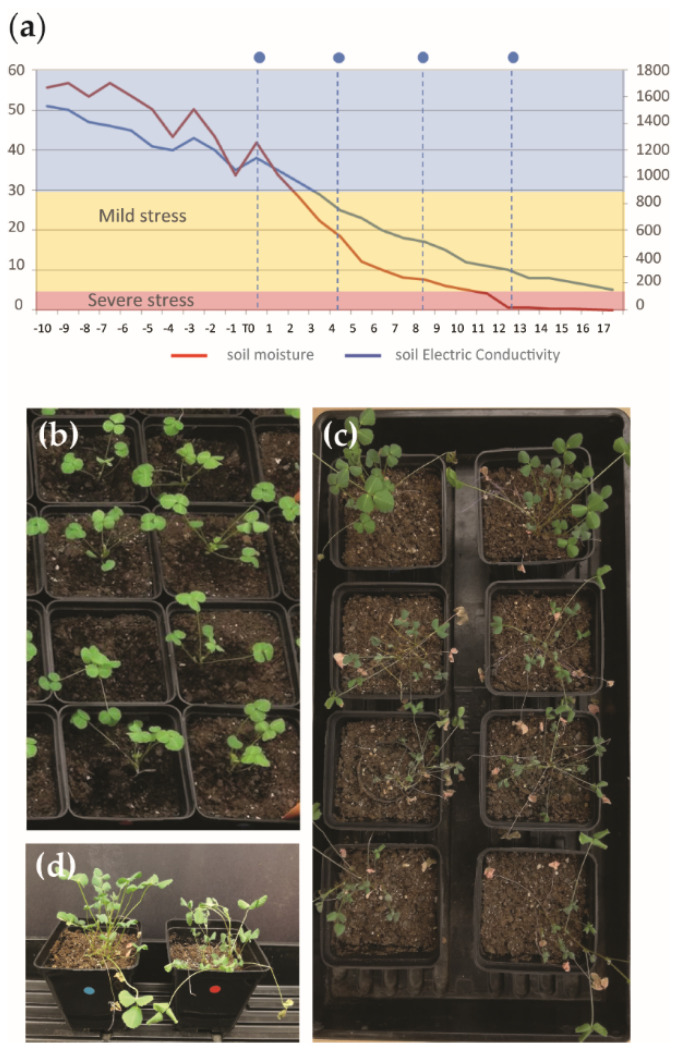
*MtKNOX3-like* RNAi transgenic plants display reduced tolerance to drought stress. (**a**) Set up of drought stress experiment and monitoring of water stress condition over time: soil moisture (blue line) and soil electric conductivity (red line). The degree of water stress was indicated in yellow (mild) or pink (severe) following the parameters from Zhang et al. 2014 [29]. Blue dots and dotted lines indicated the time course used for the molecular and physiological analysis. (**b**) Plants from control T2 lines (CTR) (upper two plants in the Figure), or T2 derived from the three selected independent RNAi (CI38, CI73; and CI76) at the beginning of the drought stress experiment (T0) showing the same healthy phenotype. (**c**) Control (CTR) and RNAi (RNAi) T2 plant phenotype at day 12, corresponding to mild stress. (**d**) Plants from control T2 lines (CTR) (upper two plants in the Figure), or T2 derived from the three selected independent RNAi (CI38, CI73; and CI76) after 18 days of drought stress, with the latter showing a severe drought stress phenotype.

**Figure 4 ijms-24-12668-f004:**
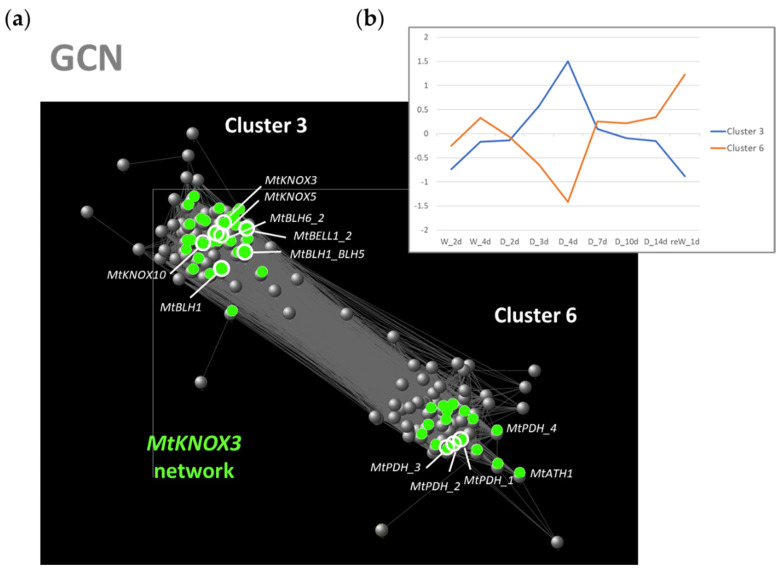
Gene co-expression network constructed using genes in Cluster 3 and Cluster 6. (**a**) Cytoscape representation of the GCN: spheres represent genes, grey edges represent connections, and green spheres represent the subnetwork formed by the *MtKNOX3-like* genes. Labels represent the TALE (KNOX and BELL/BLH) genes involved in the GCN, and the main hubs in the network. (**b**) Representation of the centroids of Cluster 3 (blue line) and Cluster 6 (orange line) that represent the average gene expression within a cluster across all the probes in the analysis. W_2d = watered plants at day 2; W_4d = watered plants at day 4; D_2d = drought stress at day 2; D_3d = drought stress at day 3; D_4d = drought stress at day 4; D_7d = drought stress at day 7; D_10d = drought stress at day 10; D_14d = drought stress at day 14; reW_1d = plants re-watered for 1 day after drought stress [29].

**Figure 5 ijms-24-12668-f005:**
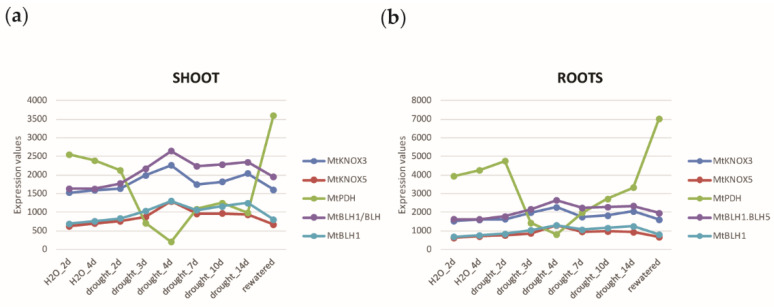
Line graph of *KNOX/BLH* gene expression from the shoot and roots transcriptomic data from Zhang et al. 2014 [29]. (**a**) *KNOX/BLH* and *MtPDH* gene expression in the shoot under drought stress; and (**b**) *KNOX/BLH* and *MtPDH* gene expression in the roots under drought stress.

**Figure 6 ijms-24-12668-f006:**
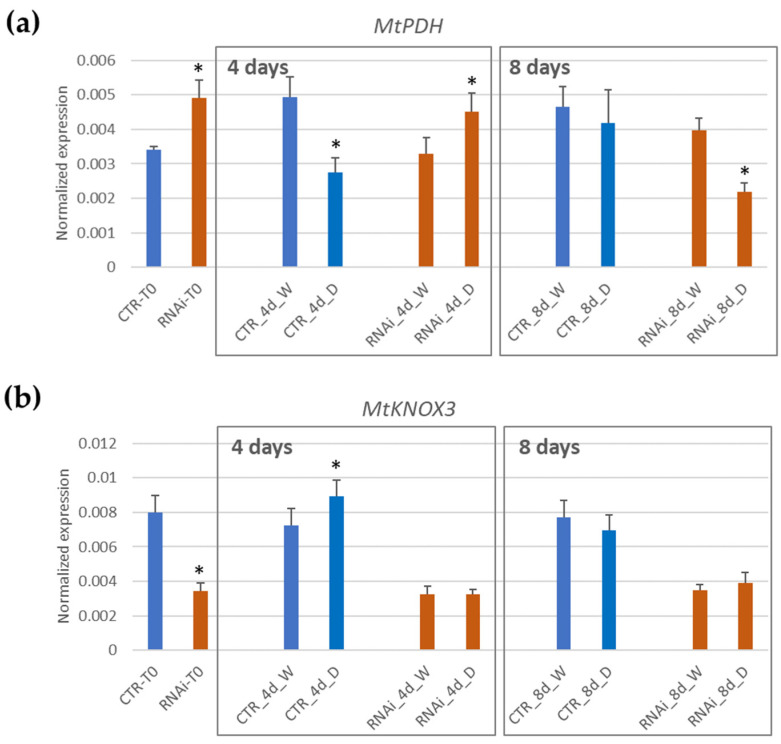
Quantitative real-time PCR analysis of *MtKNOX3* (**a**) or *MtPDH* (**b**) expressions in the leaves of a representative RNAi T2 transgenic line (RNAi) from well-watered plants (W) or drought-stressed (D), compared to a control line (CTR), at 4 days or 8 days of treatment. In each graph, the asterisk (*) indicates a statistically significant difference (Student’s *t*-test: *p* ≤ 0.05) related to watered plants.

**Figure 7 ijms-24-12668-f007:**
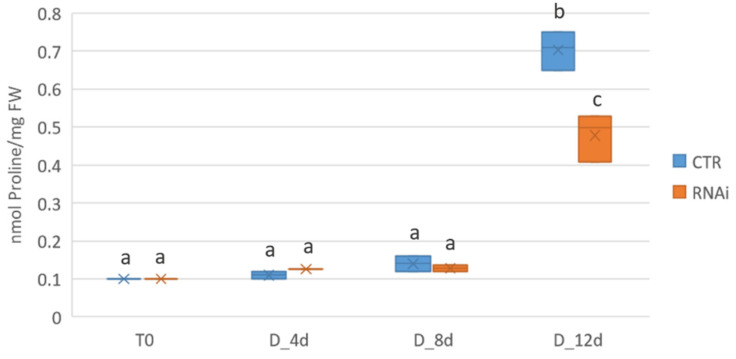
Proline accumulation in the leaves of either the control (CTR) and *Mt KNAT3/4/5-like RNAi* lines (RNAi) at 4, 8, and 12 days of drought stress (D), respectively. The means of three independent RNAi lines from two plants for each biological replicate are shown. In each graph, means with different letters represent significant difference at *p* ≤ 0.05 (Tukey’s test).

**Figure 8 ijms-24-12668-f008:**
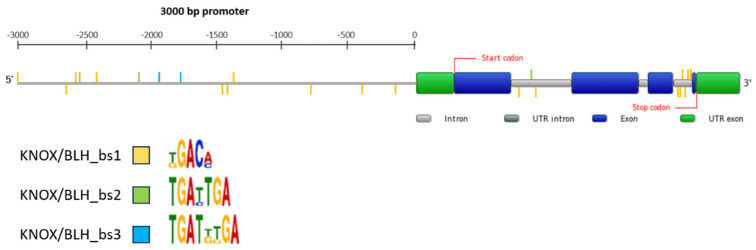
Localization of the KNOX/BLH putative binding sites in the promoter and intronic regions of the *MtPDH* gene. The gene structure of *MtPDH* was retrieved by the Plaza 5.0 web site [45] and modified with promoter information. Colored lines along the gene indicate the binding sites’ localization.

## Data Availability

All data are contained within the article or available as Appendix A.

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
