# Peer review of "A Novel Role of Medicago truncatula KNAT3/4/5-like Class 2 KNOX Transcription Factors in Drought Stress Tolerance"

_ijms, 2023, doi:10.3390/ijms241612668_

Round 1

Reviewer 1 Report

The paper by Iannelli et al. presents data showing the role of the KNAT3/4/5 class 2 KNOX transcription factors in drought stress tolerance in Medicago truncatula. This is an interesting result that deserve publication.

The paper is well written with only a mistake in Figure 2. In the text, line 204, there is a citation for figure 2d, but the panel d is not indicated on the figure and not mentioned in the legend. This should be corrected.

My main concern is that the paper looks like two stories that were put together. The reasoning in the discussion explains better the paper (question, bioinfo search and experiment to validate the reasoning). I mean that the story as it is given does not seems the more logical. The authors should first give the bioinformatic approach, using the previous Arabidopsis results as the reason to do the search, validate the gene networking with the qRT-PCR and then use the transgenic to validate the story. This means order the paragraph differently like paragraphs 2.4/2.5/2.1/2.2/2.3/2.6. I believe this would make the story more linear and like a “story”. Part of the introduction might be rewritten to fit to this.

In addition an easy addition to the paper could be to explore the MtPDH promoter for KNOX/BLH binding sites to support their conclusions. Ideally activation of the promoter in a heterologous system (Nicotiana ?) could also be possible but it will require additional experiments. Any additional results that could show the interaction of the TF with the promoter could be valuable.

Author Response

We thank the reviewer for helpful and appropriate comments and suggestions. Indeed, the study can be seen as three parts: on one side we used a classical reverse genetic approach to study the role of MtKNOX3-like genes and found that plants with reduced expression of MtKNOX3-like genes (RNAi) are less tolerant to drought; then we predicted possible targets that may explain this phenotype by applying a home-made bioinformatic approach to M. truncatula transcriptomic data that were previously published by another research group (Udvardi lab). Finally, we went back to our transgenic plants and validated the bioinformatic prediction in planta showing that the best target candidate gene, MtPDH gene, is indeed deregulated in the RNAi plants both in controlled or stress conditions. We agree that the “story” would be more appealing starting from the bioinformatic approach. However, this is the real story of this study, and the transcriptomic data that we analyzed were produced, and kindly made available to the community, by other colleagues (Udvardi lab). We feel that it would not be fair to change the order of the experiments and start from the data produced by others, although the idea was excellent.

Another excellent point would be to establish whether the MtKNOX3-like TFs regulation of MtPDH is direct or indirect. As you observed, this would require additional experiments thus delaying publication and endangering the novelty of the discovery. We therefore responded to your request of exploring the MtPDH promoter for KNOX/BLH binding sites. For the analysis we used information from analyses made for KNOX1 and BLH binding site, as no binding site specificity information is available for KNOX2 genes of KNAT3/4/5 clade. Our analysis was added to the Results section of the manuscript (Figure 8, Supplementary Figure S2, Supplementary Table S8). Fourteen putative KNOX/BLH binding sites were found in the 3000bp upstream regions of the MtPDH gene and nine in the intronic sequences. In particular, the third intron (187 bp length) of the MtPDH gene was strikingly enriched in KNOX/BLH binding sites (6 binding sites) and may constitute a main regulatory element that could be tested in further analysis to validate MtKNOX3-like TFs and MtPDH regulatory relationship.

Reviewer 2 Report

I suggest this paper for publication. It is a high-quality manuscript using recently developed analysis techniques.

Author Response

We thank the Reviewer for appreciating our manuscript.

Reviewer 3 Report

The manuscript entitled “A novel role of Medicago truncatula KNAT3/4/5-like class 2 KNOX transcription factors in drought stress tolerance” described the functional of KNAT3/4/5-like class 2 KNOX transcription factors in drought stress tolerance. The author produced transgenic M. truncatula(RNAi lines) and also measured the leaf water loss, leaf wilting time, and chlorophyll loss after drought treatment of RNAi lines and Control lines. In addition, the author reanalyzed a transcriptomic M. truncatula drought stress experiment and pointed to a possible role of MtKNOX3-like transcription factors in repressing a proline dehydrogenase gene (MtPDH) and inhibiting proline degradation. However, the flowing points should be addressed before considering acceptance for publishing in the International Journal of Molecular Science.

Major concerns

1.     In the whole experimental design, the author did not produce the single gene (KNOX 3, 5, 9 or 10) RNAi lines, double mutant, and triple mutant, otherwise it cannot conclude which KNOX2 gene functions in drought response. Although they are functional redundancy, the author should show the phenotype of the above RNAi plants to confirm that. 

2.     At the presence of the first question, they can not confirm which KNOX 2 gene (KNOX 3, 5, 9 10) targets the downstream genes to control their transcription because TFs regulate the transcription of target genes to control pants growth, development, and stress response.

3.     Meanwhile, the author concludes that the KNOX3-like genes inhibit proline degradation through the regulation of the MtPDH gene. Here, you should provide more evidence such as Y1H, EMSA, and Dual LUC assay to confirm the combination and inhibition of KNOX2 (Which one) on MtPDH gene.

4.     All figures are unclear with low resolution. It is strongly recommended that the author should use the standard software to plot. For instance, Origin, GraphPad, or R. The style of all symbols in the figure should be in one style. Please detailed the figure legends, the notes of some numbers or symbols are lost. Eg. In Fig.1, CI138, Ci140…… and P1, P23……

5.     The author should provide the phenotype before drought and after re-watering, not only the phenotype after stress in Figure 3.

6.     Please provide all data of CI38, 73, and 76 lines instead of RNAi in figures 2, 3(b), 6, and 7.

7.     Two biological replicates are not enough. There must be three biological replicates at least.

Minor concerns

Abstract

Line 19 please simply the sentence.

Line 22 the results of leaf water loss, leaf wilting time, and chlorophyll loss are unclear

Introduction

1.     Throughout the articles, including lines 14,73,84,132,418,419,426,713,714, Arabidopsis should be italic. Please check this through the whole manuscript.

2.     Line 64 transcription factors should be TFs

3.     Line 76-77, “although there are some indirect indications that this could be the case”. Is there any reference that can support this inference? please cite the reference.

Results

1.     In Figure 1, please provide a picture of the RNAi lines, not only RNAi hox plants.

2.     Line 199, “leaf senescence assay at different time points (0d, 6d and 10d)”. The time point in parentheses should be corrected to 0d, 3d, 4d, 5d and 6d.

3.     Line 204, “Figure 2d” should be corrected to Figure 2c. There is no Figure 2d in the figure 2.

4.     Line 342, “transcriptional repressors (ref.)”. Missing reference here.

Method

1.     Line500, Why was R108 genotype selected instead of A17 for genetic transformation?

Discussion

If some of the above concerns could not be addressed, the author should discuss them in this part and provide reasonable explain.

no

Author Response

The manuscript entitled “A novel role of Medicago truncatula KNAT3/4/5-like class 2 KNOX transcription factors in drought stress tolerance” described the functional of KNAT3/4/5-like class 2 KNOX transcription factors in drought stress tolerance. The author produced transgenic M. truncatula (RNAi lines) and also measured the leaf water loss, leaf wilting time, and chlorophyll loss after drought treatment of RNAi lines and Control lines. In addition, the author reanalyzed a transcriptomic M. truncatula drought stress experiment and pointed to a possible role of MtKNOX3-like transcription factors in repressing a proline dehydrogenase gene (MtPDH) and inhibiting proline degradation. However, the flowing points should be addressed before considering acceptance for publishing in the International Journal of Molecular Science.

Response to reviewer 3  

We thank the reviewer for helpful and appropriate comments and suggestions. We addressed reviewer’s concerns and criticisms. Please find below our point-by-point response.

Major concerns

  1. In the whole experimental design, the author did not produce the single gene (KNOX 3, 5, 9 or 10) RNAi lines, double mutant, and triple mutant, otherwise it cannot conclude which KNOX2 gene functions in drought response. Although they are functional redundancy, the author should show the phenotype of the above RNAi plants to confirm that.

Response: The very similar expression pattern of KNAT3/4/5-like genes during plant development in both Arabidopsis and M. truncatula points to a functional redundancy. Lack of phenotypes was already reported for Arabidopsis KNAT3/4/5 single mutants in which the expression of the single genes had been knocked out (Furumizu et al. 2015). Nevertheless, we did produce some RNAi transgenic lines where the most abundant KNOX2 gene, MtKNOX3, was knocked down via RNAi (data not shown), but no drought response or other evident phenotypes were observed. In our previous work on the role of KNAT3/4/5-like genes in M. truncatula nodule symbiosis we observed no phenotype when only MtKNOX3 was knocked down (Di Giacomo et al. 2017). Moreover, we observed that downregulating MtKNOX3 would result in overexpression of MtKNOX5, and overexpression of MtKNOX3 associated with a reduced expression of MtKNOX9 and MtKNOX10, thus suggesting strong transcriptional feedback regulation amongst the KNAT3/4/5-like genes (Di Giacomo et al. 2017). These previous findings convinced us that only producing plants in which all the four genes were knocked down could give us the possibility to acquire functional information. For clarity, we added these considerations in the manuscript at the beginning of the Discussion section.

  1. At the presence of the first question, they can not confirm which KNOX 2 gene (KNOX 3, 5, 9 10) targets the downstream genes to control their transcription because TFs regulate the transcription of target genes to control pants growth, development, and stress response.

Response:  As explained above, the level of redundancy of this KNOX2 clade is very high, and functional studies have been limited by the high degree of redundancy amongst members of this gene family (Truernit & Haseloff, 2007). KNAT3, KNAT4, and KNAT5 were shown to act redundantly in regulating leaf morphology in Arabidopsis and leaf complexity in Cardamine hirsuta (Furumizu et al., 2015). No specific target genes have been identified so far for the single members of this clade. These genes are largely co-expressed and the encoded proteins heterodimerize with BELL (BLH) TALE homeobox proteins to form functional complexes in a combinatorial manner. The transcriptomic data that we analysed in our manuscript also indicated a strong co-expression and co-regulation of MtKNOX3-like genes, and identified several putative co-regulated BLH partners. The function and target genes of these several interacting TFs are already difficult to identify and predict, even more difficult to pinpoint specific functions within MtKNOX3-like clade. KNOX TFs may need the specific BLH partner to bind and regulate downstream target genes, as binding specificity and function depends on KNOX/BLH specific interactions (Kim et al. 2013; Di Giacomo et al. 2013). We believe that our findings about the MtPDH gene as a putative target of the KNOX/BLH TF co-expressed in Cluster 3 is one of the few reports about the identification of KNOX2 downstream regulated genes, and the first suggesting a role of KNOX2 in drought response. Although further analysis would be needed to clarify the biochemical and molecular mechanisms of this possible regulation, we believe that our data are novel and any further delay would endanger the novelty of the discovery.

  1. Meanwhile, the author concludes that the KNOX3-like genes inhibit proline degradation through the regulation of the MtPDH gene. Here, you should provide more evidence such as Y1H, EMSA, and Dual LUC assay to confirm the combination and inhibition of KNOX2 (Which one) on MtPDH gene.

Response:  As previously explained, to provide more evidence regarding the direct regulation of MtPDH gene by MtKNOX3-like TFs, and which one, is not an easy task as different combinations of KNOX/BLH should be tested to achieve this result, and goes beyond the scope of our manuscript. Moreover, these experiments would be quite time consuming, thus delaying publication and endangering the novelty of the discovery. However, as asked by Reviewer 1, we added a novel analysis in the manuscript to corroborate our regulatory hypothesis and explored the MtPDH promoter for KNOX/BLH binding sites. For the analysis we used information from the analyses made for KNOX1 and BLH binding site, as no binding site specificity information is available for KNOX2 genes of KNAT3/4/5 clade. Our analysis was added to the Results section of the manuscript (Figure 7, Supplementary Figure S2, Supplementary Table S8). Fourteen putative KNOX/BLH binding sites were found in the 3000bp upstream regions of the MtPDH gene and nine in the intronic sequences. In particular, the third intron (187 bp length) of the MtPDH gene was strikingly enriched in KNOX/BLH binding sites (6-fold with respect to the expected motif frequency) and may constitute an important cis-element in the MtKNOX3-like TF and MtPDH regulatory relationship.

  1. All figures are unclear with low resolution. It is strongly recommended that the author should use the standard software to plot. For instance, Origin, GraphPad, or R. The style of all symbols in the figure should be in one style. Please detailed the figure legends, the notes of some numbers or symbols are lost. Eg. In Fig.1, CI138, Ci140…… and P1, P23……

Response: we checked the quality of the Figures and when needed we improved their quality (numbers) or changed them according to Reviewer’s suggestions (Figure 1, Figure 2, Figure 3, Figure S1…..).

  1. The author should provide the phenotype before drought and after re-watering, not only the phenotype after stress in Figure 3.

Response: the phenotype of plants before drought was included in Figure 3. Regarding rewatering, we could not recover transgenic plants by rewatering after 18 days of drought. We added this details in the manuscript.

  1. Please provide all data of CI38, 73, and 76 lines instead of RNAi in figures 2, 3(b), 6, and 7.

Response: The three independent transgenic T2 lines, which progeny was genotyped for each experiment as described in Mat&Met, can be considered as the biological replicates. Therefore, we believe it is more correct to have all the lines together indicated as “RNAi” in the main text.

  1. Two biological replicates are not enough. There must be three biological replicates at least.

Response: as above stated, we always used biological triplicates, i.e., three independent transgenic T2 lines, as indicated in all the legends and Mat&Met section.

Minor concerns

Abstract

Line 19 please simply the sentence.

Response: This sentence should well explain the plant material that was produced and analysed. Therefore, we cannot simplify the sentence without making it less clear.

Line 22 the results of leaf water loss, leaf wilting time, and chlorophyll loss are unclear:

Response: We changed accordingly.

Introduction

  1. Throughout the articles, including lines 14,73,84,132,418,419,426,713,714, Arabidopsis should be italic. Please check this through the whole manuscript.

Response: We changed accordingly.

  1. Line 64 transcription factors should be TFs

Response: We changed accordingly.

  1. Line 76-77, “although there are some indirect indications that this could be the case”. Is there any reference that can support this inference? please cite the reference.

Response: We changed accordingly.

Results

  1. In Figure 1, please provide a picture of the RNAi lines, not only RNAi hox plants.

Response: In Figure 1 we provided an example of RNAi hox plants because we could not retrieve any T1 progeny. The phenotype of the other transgenic RNAi lines is similar to control empty vector plants in the absence of stress can now be found in Figure 3b as a control.

  1. Line 199, “leaf senescence assay at different time points (0d, 6d and 10d)”. The time point in parentheses should be corrected to 0d, 3d, 4d, 5d and 6d.

Response: The time points in line 199 are correct as they refer to chlorophyll content measurement.

  1. Line 204, “Figure 2d” should be corrected to Figure 2c. There is no Figure 2d in the figure 2.

Response: We changed accordingly.

  1. Line 342, “transcriptional repressors (ref.)”. Missing reference here.

Response: We added the reference.

Method

  1. Line500, Why was R108 genotype selected instead of A17 for genetic transformation?

Response: The R108 ecotype has a much higher transformation efficiency as compared to the A17 ecotype and it is commonly used for stable transformation.

Discussion

If some of the above concerns could not be addressed, the author should discuss them in this part and provide reasonable explain.

Response: We tried to address all the concerns providing reasonable explanations through point-by-point responses, with particular attention to flowing points 1, 2 and 3.

Reviewer 4 Report

Manuscript ID: ijms-2506448v1

Title:  A novel role of Medicago truncatula KNAT3/4/5-like class 2 KNOX transcription factors in drought stress tolerance

Authors: Iannelli MA et al.

General comments: The manuscript investigates the role of class 2 KNOX like TFs in regulation of drought tolerance. However, the major focus of the study is only anti-correlation of KNOX3 and PDH at only 4D time point after drought stress. The study uses a large amount of transcriptomic data for gene network analysis and co-expression patters and focused on one pair, KNOX3 and PDH. Several other factors should have been addressed to conclude for the major concluding statements of the study. Drought is a complex stress, and its response is also very complex, and mediated by several factors operating at multiple levels. It is difficult to associate the role of KNOX3 proposed based on the limited experimental evidences in the study, including the data of RNAi lines vis-à-vis control plants. Some of the comments are detailed below, and also indicated the PDF file.

Section specific comments

Title:                       The title apparently looks fine but the novelty highlighted needs to be visible in the data presented. Some times that link of regulation KNOX3 mediated PDH regulation looks missing. Hence it may need revision after the concerns raised below are addressed

Abstract:                The abstract looks fine, and few minor comments are indicated in the PDF file. However, it may need some modifications in view of the over all comments.

Introduction:        The introduction section is very long and may be written in a concise manner. Minor changes have been indicated in the PDF file, and some concerns are listed below:

Lines 33-38:           May need rewriting for better clarity.

Line 74:                  Cite publication in support.

Lines 88-90:           In drought adapted species it may be possible that certain genes/mechanisms may have been well optimized for higher basal levels and may not exhibit induction like other plants?

Line 102-125:        This paragraph may be written in a concise manner.

Line 135-141:        Is the role of these KNOS-3 like TFS in negative PDH regulation shown for the first time?

Results:                  This section needs considerable improvement, by removal of repetitive description at several places as in materials & methods section, and appropriate data description in text w.r.t. results evident in the figures. Also, statements with previous citations may be removed/minimized and restricted to introduction/discussion.

Lines 150-151:      Which of two category of plants develop seeds may be indicated, as the Mt KNAT3/4/5-like hox RNAi  died as mentioned in next statement.

Lines 179-181:      Significance should be indicated for the plants in the Figure?   

Lines 192-195:      Such content may be removed from results, and may be moved to M&M or discussion, if needed.  

Line 199:                Fig 2C should be on the left side panel in the Figure 2. As of now the Figure 2C and 2D are placed in a reverse manner w.r.t. to their description in the text and Figure 2D label is missing.

Lines 201-202:      What impact will this have on all other characteristics of the RNAi plants analyzed. The rate of reduction of total chlorophyll may be compared.

Lines 202-203:      This statement should be supported by statistical significance data.

Lines 204-206:      This statement may be moved to the discussion part.

Lines 224-234:      This detailed description may be moved to the M&M section, and if already there this may be removed/minimized, and directly results may be discussed.

Lines 254-255:      Description under this section may be written in a concise manner., and some statements that are more appropriate for materials and methods section may be moved to that section.

Lines 256-259:      Some of the description is appropriate for M&M section. How much analysis of drought-induced gene expression has already been reported in the previous high-throughput analysis? Does the expression pattern or co-expression networks of the genes analyzed in the present study also reported in these studies? This must be indicated in the introduction/discussion.

Line 260-263:        This description may be moved to the M&M section.

Lines 264-271:      Some statements may be moved to M&M section and only results cane be part of this section.

Lines 271-278:      There are many other genes also that showed up-regulation at 4D time-point, This also see in K1 cluster?

Lines 288-290:      And post 4D time-point down-regulation was also observed in clusters K7 and K10? Why these were excluded from analysis as these down-regulation can also be little bite out-of phase (delayed) than the up-regulated genes in K3 cluster at 4D time-point? What types of changes in 4-D drought stressed shoots lead to selection of Cluster 3 and 6 genes, as Cluster 1/9 and 7/10 also showed inverse correlation after drought stress?

Line 290-293:        This statement may be moved to M&M section.

Lines 293-294:      Repetitive statement as above, may be removed.

Lines 296-300:      This may also be moved to the M&M section, ans some component can be part of introduction and/or discussion.

Lines 319-325:      Some details may be moved to the M&M section.

Line 330:                The results should also show if proline was accumulated in the plants under conditions used in this study and what was the pattern of P5CS genes (directly involved in proline biosynthesis). The overall proline levels are outcome of dynamics of p5CS and PDH genes. This may also be checked.

Lines 342-343:      Are there any reports, indicating direct interaction of KNOX2 TFs with regulatory regions of these target genes? How many other regulators of PDH are known, which are involved in their up/down-regulation in different conditions? Whats are their patterns in this study? Is  the PDH down-regulation solely attributed to KNOX2 factors, appropriate?

Lines 346-347:      This statement with a previous report citation may be moved to Discussion section.

Lines 348-350;      Roots experience the stress first followed by the upper parts of the plant. Why the expression profiles very similar in the two plant parts on day 4 of stress? Are similar observation also reported previously for other genes?

Lines 351-353:      Why the expression patterns returns to more or less normal after 4D time-point and on re-watering it jumps sever folds?

Lines 360-361:      Cluster 1/9 and 7/10 also represent the same behavior, and actually a more continuous inverse expression correlation (up/down) pattern? Why these were exclude in favor of Cluster 3 and 6 is not clearly evident and may need better explanation in the manuscript.

Lines 361-364:      This experiment may only tell about the expression levels and not if the KNOX3 TFs directly/indirectly regulates the PDH expression or not. There are many other positive and negative regulators of PDH expression, the roles of which cannot be ruled out in favor of KNOX3s.

Lines 365-368:      But how increased PDH will contribute to the enhanced proline levels is not clear. and Teh KNOX3 is not showing any difference between watered and drought stressed plants at 4D as well as 8D? this is not clear?

Lines 368-369:      At both 4D and 8D the levels of KNOX is similar, then why the PDH levels are different (higher or lower). This indicates that there are other key factors also, in addition to KNOX3.

Lines 370-372:      Even if this is as explained, why the KNOX3 down-regulation is not able to regulate the levels of PDH at other points? Other regulators may also be involved and cannot be ruled out. Even at early stages of drought, apart from KNOX3 what other factors regulate PDH is not investigated. Also the role of proline biosynthesis genes needs to be analyzed before suggesting that KNOX3 mediated PDH regulation is the key to drought stress?

Lines 379-383:      The letters are same (a) from T0 to 8D between the two plants, indicating no significant difference? How then the proline accumulation can be visualized in different between the two lines and mediated by KNOX3? Hence role of other three proline biosynthesis genes cannot be excluded, as they are also induced in different stresses including drought.

Lines 389-390:      The letters are same (a) from T0 to 4D between the two plants, indicating no significant difference?

Lines 390-391:      The letters are same (a) from 8D indicating no difference between the two plants., However only at 12D the difference seems significant. The analysis in the manuscript mostly focused on 4D time-point.

Lines 394-395:      This will need more experimental support as indicated in the previous queries.

Discussion:            Discussion section will also need to be updated based on the comments in the results, may be written in a concise manner.

Lines 401-409:      These statements are similar to results, may be minimized or removed.

Lines 411-420:      Content same as in Introduction may be removed or minimized.

Lines 432-435:      The basis of this selection is not clear, as the drought stress is continuously increasing, and there were two other cluster groups with associated up/down regulation trend.

Lines 441-446:      Description may be reduced.

Lines 447-449;      The trend the proline biosynthesis genes should have been evaluated along with PDH.

Lines 456-464: This is important aspects as P5CS roles in proline accumulation has not been addressed in the results, and the proline accumulation cannot be solely attributed to the PDH regulated by KNOX3, which is also not a continuous phenomenon throughout the stress.

Lines 472-473:      The RNAi lines accumulate similar amount of proline till 8D and there is not statistical difference. Kindly recheck?

Lines 484-486: Some indicators of oxidative stress should have been analyzed also.

Material and methods: Comments are given below and also in the PDF file.

Lines 505-506:      Humidity conditions may also be mentioned. Kindly check the units of PFD. the commonly used unit is μmol/m2/s.

Lines 535-536:      Give conc. in molarity (milli or micro), Indicate Units of enzyme.

Line 540:                This section may be combined with section 4.1, by appropriately modifying the title. Some repetitive content may be removed.

Line 544:                Kindly check the units of PFD. the commonly used unit is μmol/m2/s.

Line 556:                The protocols described under this section may be written in a concise manner and by citing previous publications at appropriate places.

Line 596:                What Arabidopsis plants are mentioned here, these are not used in this experiment?

Lines 606-608:      If methods form a previous report is followed kindly cite it.

Figures: The concerns related to the Figures are indicated in the PDF file and some are listed below.

Figure 1: The significance symbols are not indicated in the Fig 1 C and D panels.

Figure 2: Figure 2C should be on left side while 2d should be moved to right side and label of 2d missing. Text indicates Fig 2c and 2d, kindly rectify if needed

Figure 3: Detailed legend description may be moved to the M&M section. Some statements are repetitive as in results as above, It may be removed or description modified.

Figure 4: The details for GCN analysis are mentioned at several places. This can be minimized and  can be in M&M.

Figure 5: Kindly check if this data/ figure may need permission from the Authors/Publisher of 2014 paper.

Figure 6: IF the levels of PDH are compared in the RNAi lines at T0 and T4 there is not difference although it reduces in watered conditions. The comparisons needs to be done carefully. Why PDH is not getting down-regulated in control plants subjected to drought after 4D time-point. Statistical significance at 4D time-point in control W and Control drought looks very low or appears insignificant (although marked at *). The method used for analysis and actual p values may be indicated)

Figure 7: The letters are same (a) from T0 to 8D between the two plants, indicating no significant difference? How then the proline accumulation can be visualized in different between the two lines and mediated by KNOX3? Hence role of other three proline biosynthesis genes cannot be excluded, as they are also induced in different stresses including drought.

Supplementary Data: Supplementary data files are fine but it should be made sure that the captions or legends should be sufficiently descriptive for easy understanding.

Quality of English language used in the manuscript is fine. Minor changes may be needed at few places.

Author Response

Reviewer 4

Manuscript ID: ijms-2506448v1

Title:  A novel role of Medicago truncatula KNAT3/4/5-like class 2 KNOX transcription factors in drought stress tolerance

Authors: Iannelli MA et al.

General comments: The manuscript investigates the role of class 2 KNOX like TFs in regulation of drought tolerance. However, the major focus of the study is only anti-correlation of KNOX3 and PDH at only 4D time point after drought stress. The study uses a large amount of transcriptomic data for gene network analysis and co-expression patters and focused on one pair, KNOX3 and PDH. Several other factors should have been addressed to conclude for the major concluding statements of the study. Drought is a complex stress, and its response is also very complex, and mediated by several factors operating at multiple levels. It is difficult to associate the role of KNOX3 proposed based on the limited experimental evidences in the study, including the data of RNAi lines vis-à-vis control plants. Some of the comments are detailed below, and also indicated the PDF file.

Response: We thank the reviewer for helpful and appropriate comments and suggestions that allowed us to consistently improve the quality of our manuscript.

Section specific comments

Title:                       The title apparently looks fine but the novelty highlighted needs to be visible in the data presented. Some times that link of regulation KNOX3 mediated PDH regulation looks missing. Hence it may need revision after the concerns raised below are addressed

Response: We believe that the Title reflects the data presented and the novelty since a role of KNOX2 TFs in drought stress was not described so far.

Abstract:                The abstract looks fine, and few minor comments are indicated in the PDF file. However, it may need some modifications in view of the over all comments.

Response: We addressed all the minor comments according to reviewer’s suggestions

Introduction:        The introduction section is very long and may be written in a concise manner. Minor changes have been indicated in the PDF file, and some concerns are listed below:

Lines 33-38:           May need rewriting for better clarity.

Response: We have revised accordingly.

Line 74:                  Cite publication in support.

Response: We added the citation.

Lines 88-90:           In drought adapted species it may be possible that certain genes/mechanisms may have been well optimized for higher basal levels and may not exhibit induction like other plants?

Response: Yes, this could be one of the many possible mechanisms to adapt to drought.

Line 102-125:        This paragraph may be written in a concise manner.

Response: We shortened the paragraph to make it more concise.

Line 135-141:        Is the role of these KNOS-3 like TFS in negative PDH regulation shown for the first time?

Response: Yes, this is the first time that a role for KNOX3-like genes in regulating a key drought stress gene such as PDH, and in general stress response, is described. No specific target genes have been identified so far for these TFs. We added this information in the text at lines XXX.

Results:                  This section needs considerable improvement, by removal of repetitive description at several places as in materials & methods section, and appropriate data description in text w.r.t. results evident in the figures. Also, statements with previous citations may be removed/minimized and restricted to introduction/discussion.

Response: We shortened this section according to your and other reviewer’s suggestions.

Lines 150-151:      Which of two category of plants develop seeds may be indicated, as the Mt KNAT3/4/5-like hox RNAi  died as mentioned in next statement.

Response: We specified the category of plants developing seed in the text.

Lines 179-181:      Significance should be indicated for the plants in the Figure?   

Response: We did add significance in the Figure.

Lines 192-195:      Such content may be removed from results, and may be moved to M&M or discussion, if needed.  

Response: We removed such content and moved to M&M.

Line 199:                Fig 2C should be on the left side panel in the Figure 2. As of now the Figure 2C and 2D are placed in a reverse manner w.r.t. to their description in the text and Figure 2D label is missing.

Response: We corrected the order of the panels in Figure 2.

Lines 201-202:      What impact will this have on all other characteristics of the RNAi plants analyzed. The rate of reduction of total chlorophyll may be compared.

Response: We changed the panel and indicated the rate of reduction of total chlorophyll and the statistical significance.

Lines 202-203:      This statement should be supported by statistical significance data.

Response: We added significance data.

Lines 204-206:      This statement may be moved to the discussion part.

Response: We believe that this statement can help readers to interpret the importance of the biochemical results. To put it in the discussion would imply to recall the biochemical experiment, thus lengthening the Discussion part.

Lines 224-234:      This detailed description may be moved to the M&M section, and if already there this may be removed/minimized, and directly results may be discussed.

Response: We removed such content and moved to M&M.

Lines 254-255:      Description under this section may be written in a concise manner., and some statements that are more appropriate for materials and methods section may be moved to that section.

Response: We have revised accordingly.

Lines 256-259:      Some of the description is appropriate for M&M section. How much analysis of drought-induced gene expression has already been reported in the previous high-throughput analysis? Does the expression pattern or co-expression networks of the genes analyzed in the present study also reported in these studies? This must be indicated in the introduction/discussion.

Response: Regarding the description of the experiment done in Zhang et al. we cannot describe it in our M&M. A brief introduction of the study is necessary for the readers to understand the further analysis presented in the manuscript.

With regard to the bioinformatic analysis done in the study of Zhang et al. 2014, they identified many genes which expression is tightly coupled to the plant water potential. Combining M. truncatula metabolomic data with transcriptomic data they gained new insight into the regulation of metabolic pathways operating under drought stress. Among the metabolites detected in drought-stressed plants, they observed that proline had striking regulatory profiles indicating involvement in M. truncatula drought tolerance. What is new in our analysis is that we used a knowledge-based targeted approach to focus specifically on TFs and few selected pathways of drought response. A subset of transcripts was used for cluster analysis and gene co-expression networks that allowed us to improve the sensitivity of the analysis in order to specifically identify genes and pathways associated with our TF class of interest, the MtKNOX3-like TFs. The expression pattern of these genes was not reported in that study. We added these considerations in both Introduction and Discussion.

Line 260-263:        This description may be moved to the M&M section.

Response: Regarding the description of the Affymetrics system, we believe that it is important to properly describe it to make readers understand the further analysis presented in the manuscript.

Lines 264-271:      Some statements may be moved to M&M section and only results cane be part of this section.

Response: We moved some content to M&M.

Lines 271-278:      There are many other genes also that showed up-regulation at 4D time-point, This also see in K1 cluster?

Response: It is true, there are many other genes that are upregulated at 4D time-point in cluster K1, that represent those genes that steadily increase during drought stress, and have been studied in Zhang et al. However, this Cluster is not correlated with the expression pattern of our genes of interest (in K3) and therefore we were not interested in it.

Lines 288-290:      And post 4D time-point down-regulation was also observed in clusters K7 and K10? Why these were excluded from analysis as these down-regulation can also be little bite out-of phase (delayed) than the up-regulated genes in K3 cluster at 4D time-point? What types of changes in 4-D drought stressed shoots lead to selection of Cluster 3 and 6 genes, as Cluster 1/9 and 7/10 also showed inverse correlation after drought stress?

Response: This is also true, but these clusters are not related with the pattern of expression of our genes of interest. In our study we only addressed genes either co-regulated or anti-correlated with KNOX3-like genes in order to characterize their role in drought stress response.

We focused on the cluster were most of MtKNOX3-like genes placed (Cluster 3), and on the anti-correlated one (Cluster 6) to predict possible target genes. The other clusters may be more interesting for studying drought response, and were considered in the study of Zhang et al 2014, but our interest was solely to use this excellent transcriptomic experiment to predict possible targets of the TF of our interest that could explain decreased tolerance of RNAi lines. We added this information in the main text to better clarity our bioinformatics approach.

Line 290-293:        This statement may be moved to M&M section.

Response: We believe that it is important to properly describe how many samples we analysed so to make readers understand the further analysis presented in the manuscript.

Lines 293-294:      Repetitive statement as above, may be removed.

Response: We removed this statement.

Lines 296-300:      This may also be moved to the M&M section, ans some component can be part of introduction and/or discussion.

Response: We partly removed these contents, as some statements are important for the readers to follow our approach.

Lines 319-325:      Some details may be moved to the M&M section.

Response: We partly removed these contents, as some statements are important for the readers to follow our approach.

Line 330:                The results should also show if proline was accumulated in the plants under conditions used in this study and what was the pattern of P5CS genes (directly involved in proline biosynthesis). The overall proline levels are outcome of dynamics of p5CS and PDH genes. This may also be checked.

Response: In the study of Zhang et al it was shown that in M. truncatula proline production and accumulation during drought-stress progression is tightly regulated by the up-regulation of several genes encoding P5CS biosynthesis enzymes, and concomitantly by the repression of genes coding for proline degrading enzymes such as proline dehydrogenase (PDH). P5CS genes are progressively induced during drought stress (our Cluster 1 in Figure S1) whereas PDH genes are specifically upregulated at day 4 (our Cluster 6 in Figure S1) following different transcription patterns, that we highlighted pathways in the new Figure S1 for clarity. Proline steadily accumulates in both shoots and roots from 4D on. However, the authors of this paper said that root proline content under drought stress was several folds higher than in shoots, despite the higher expression levels detected for P5CS in the shoots. All these findings suggest that proline accumulation in different organs and tissues may depend on the equilibrium amongst catabolism, biosynthesis and transport. These considerations were added to the discussion.

Lines 342-343:      Are there any reports, indicating direct interaction of KNOX2 TFs with regulatory regions of these target genes? How many other regulators of PDH are known ???????????, which are involved in their up/down-regulation in different conditions? Whats are their patterns in this study? Is the PDH down-regulation solely attributed to KNOX2 factors, appropriate?

Response: Proline metabolism is strongly connected to many cellular processes and is considered a regulatory hub of signaling pathways (Alvarez et al. 2022). For this reason, a complex expression regulation of the genes involved in proline metabolism is expected. Several bZIP TFs are known to induce PDH expression (as reviewed by Alvarez et al. 2022). Thus far, only one mechanism to negatively regulate PDH expression has been demonstrated: Veerabagu and colleagues (2014) showed that the Arabidopsis type-B response regulator 18 (ARR18) physically interacts with bZIP63 and negatively interferes with the positive transcriptional activity of bZIP63 on the PDH1 promoter. No direct negative regulators of PDH have been identified so far. We added these statements in the text at lines XXX and the related references.

Lines 346-347:      This statement with a previous report citation may be moved to Discussion section.

Lines 348-350;      Roots experience the stress first followed by the upper parts of the plant. Why the expression profiles very similar in the two plant parts on day 4 of stress? Are similar observation also reported previously for other genes?

Response: We agree with the Reviewer 4 that roots experience the stress first followed by the upper parts of the plant. Nevertheless, it is known that the signal of water-deficit sensing from root to shoot occurs at early stages of water stress (Schachtman and Goodger 2008). Then, in the late stages of drought, the drought-related transcriptional response has been already activated not only in roots but also in leaves. Our data are in agreement with this observation, since in our experiments at 4 days of water stress the root-to-shoot signalling already occurred and each organ activated its transcriptional stress response. The observation that the analyzed KNOX/BLH genes are strongly induced in both organs suggests an important role of these genes in the whole plant. We added these statements in the text at lines XXX and the related references.

Lines 351-353:      Why the expression patterns returns to more or less normal after 4D time-point and on re-watering it jumps sever folds?

Response: If this comment is referred to the behaviour of MtPDH gene, we can try to answer based on current literature. After rewatering, proline is not necessary any longer and needs to be degraded very quickly.

Lines 360-361:      Cluster 1/9 and 7/10 also represent the same behavior, and actually a more continuous inverse expression correlation (up/down) pattern? Why these were exclude in favor of Cluster 3 and 6 is not clearly evident and may need better explanation in the manuscript.

Response: Clustering gene expression data allows to identify substructures in the data, and identify groups of genes that: i) may share a biological function; ii) be under the same transcriptional control (co-regulated);  may be linked by a regulatory relationships (TF and their targets). Co-expression and anti-correlation is commonly used in bioinformatics pipelines to identify potential TF targets (doi: 10.1007/978-1-0716-1534-8_20; 10.1007/978-1-0716-1534-8_1; https://doi.org/10.1093/plphys/kiad332; 10.3390/plants11131625), as target genes are often co-expressed with TFs that positively regulate them, or anti-correlated with TFs acting as negative regulators. We added this information in the Results section 2.4.

As we stated above (see response to comment in Lines 288-290), we focused on the cluster were most of MtKNOX3-like genes placed (Cluster 3), and on the anti-correlated one (Cluster 6) to predict possible target genes. The other clusters may be more interesting for studying drought response, and were considered in the study of Zhang et al 2014, but they were not of our interest because the expression pattern of MtKNOX3-like genes was represented by Cluster 3.

Lines 361-364:      This experiment may only tell about the expression levels and not if the KNOX3 TFs directly/indirectly regulates the PDH expression or not. There are many other positive and negative regulators of PDH expression, the roles of which cannot be ruled out in favor of KNOX3s.

Response: as we stated above, we know that several bZIP TFs are known to induce PDH expression (as reviewed by Alvarez et al. 2022), and that ARR18 physically interacts with bZIP63 and negatively interferes with the positive transcriptional activity of bZIP63 on the PDH1 promoter (Veerabagu and colleagues, 2014) We agree with reviewer 4 that we cannot rule out that other regulators different from KNOX3-like transcription factors can also regulate PDH gene expression. However, it is a fact that i) the expression of three out of four MtKNOX3-like genes is highly anticorrelated with the MtPDH gene, ii) the downregulation of KNOX3-like genes in transgenic plants results in upregulation of this gene in several samples, iii) transgenic plants resulted affected in proline accumulation during drought stress and iv) many KNOX/BLH binding sites are present in the promoter of the MtPDH gene, as resulted by the new analysis added in this revised version. We think that our results allow us to consider these KNOX3-like genes as negative regulators of the MtPDH gene.

Lines 365-368:      But how increased PDH will contribute to the enhanced proline levels is not clear. and Teh KNOX3 is not showing any difference between watered and drought stressed plants at 4D as well as 8D? this is not clear?

Response: Increasing PDH levels do not enhance proline levels but strongly reduce them. The reciprocal regulation of P5CS and PDH, along with the activity of proline transporters, is important for the fine tuning of proline levels during development, and also during episodes of osmotic stress. Several studies report that plants overexpressing PDH genes display reduced levels of proline, as well plants in which PDH was reduced, display an increase in proline content (ref.). Regarding MtKNOX3 expression, we are not sure to have understood well the question, but indeed the expression slightly increases under stress at 4D whereas no differences are observed at 8D, as shown in the Figure and similar to what Zhang et al 2014 show. 

Lines 368-369:      At both 4D and 8D the levels of KNOX is similar, then why the PDH levels are different (higher or lower). This indicates that there are other key factors also, in addition to KNOX3.

Response: We agree that KNOX3-like genes may be not the only regulator of PDH (see response to Lines 361-364)

Lines 370-372:      Even if this is as explained, why the KNOX3 down-regulation is not able to regulate the levels of PDH at other points? Other regulators may also be involved and cannot be ruled out. Even at early stages of drought, apart from KNOX3 what other factors regulate PDH is not investigated. Also the role of proline biosynthesis genes needs to be analyzed before suggesting that KNOX3 mediated PDH regulation is the key to drought stress?

Response: We agree that other regulators may also be involved and cannot be ruled out. Regarding the investigation on other factors that regulate PDH, this was not the focus of our work that aimed to understand the possible role and involvement of KNOX3-like TFs in drought stress using RNAi constructs that down-regulate multiple KNOX3-like genes. Regarding the key role of PDH in drought stress, this was demonstrated in several species and discussed in the work of Zhang et al. 2014 for M. truncatula. In the latter study, through transcriptomics and metabolomic analyses, it was shown that “in M. truncatula proline production and accumulation during drought-stress progression is tightly regulated by the up-regulation of several genes encoding P5CS biosynthesis enzymes and concomitantly by the repression of genes coding for proline degrading enzymes such as proline dehydrogenase (PDH)”. P5CS genes are progressively induced during drought stress whereas PDH genes are specifically upregulated at day 4, following different transcription patterns. Our correlation and cluster analysis on Zhang et al. transcriptomic data shown that proline biosynthesis gene expression is not related with that of MtKNOX3-like genes: they start to be induced later than MtKNOX3-like genes and no correlation was found with our TFs of interest. This is the reason why we did not explore P5CS gene expression in our material, because from the bioinformatic analysis no indication of a possible regulatory relationship between MtKNOX3-like TF and P5CS was found. For clarity, we indicated the clusters to which PDH and P5CS genes belong in Figure S1.

Lines 379-383:      The letters are same (a) from T0 to 8D between the two plants, indicating no significant difference? How then the proline accumulation can be visualized in different between the two lines and mediated by KNOX3? Hence role of other three proline biosynthesis genes cannot be excluded, as they are also induced in different stresses including drought.

Response: We confirm that the same letter indicates not significant difference between samples. It is well known that proline anabolism and catabolism gene expression can be directly related to the high proline accumulation during drought stress. However, the relative contribution of the two components to proline homeostasis is difficult to be analysed as this is a dynamic system. As said before (response to lines 370-372) we did not find any evidence of a possible regulatory relationship between KNOX3-like TF and proline biosynthesis genes.

Lines 389-390:      The letters are same (a) from T0 to 4D between the two plants, indicating no significant difference?

Response: We confirm that the same letter indicates not significant difference between samples.

Lines 390-391:      The letters are same (a) from 8D indicating no difference between the two plants., However only at 12D the difference seems significant. The analysis in the manuscript mostly focused on 4D time-point.

Response:  As said before, it is well known that proline anabolism and catabolism gene expression can be directly related to the high proline accumulation during drought stress. However, the relative contribution of the two components to proline homeostasis is difficult to be analysed, and proline transport also play an important role in tissue-specific accumulation of this osmolyte. The final accumulation of proline at 12D may relate to a delay in downregulating PDH at 4D as a result of an alteration in this anabolism/catabolism equilibrium. We focus on 4D time-point because this is the time where MtKNOX3-like TF show transient gene expression induction and PDH shows transient gene expression reduction, with a Pearson correlation value < - 0.8, indicating a strong anti-correlation between the two expression patterns. MtKNOX3-like gene expression is not correlated with proline biosynthesis genes.

Lines 394-395:      This will need more experimental support as indicated in the previous queries.

Response:  We agree that this is our hypothesis and further biochemical analyses would be necessary to demonstrate a direct link. However, based on our experimental observations, this is a possible explanation for what we observed.

Discussion:            Discussion section will also need to be updated based on the comments in the results, may be written in a concise manner.

Response:  We revised the discussion taking into account your suggestions and criticisms.

Lines 401-409:      These statements are similar to results, may be minimized or removed.

Response:  We revised accordingly.

Lines 411-420:      Content same as in Introduction may be removed or minimized.

Response:  We revised accordingly.

Lines 432-435:      The basis of this selection is not clear, as the drought stress is continuously increasing, and there were two other cluster groups with associated up/down regulation trend.

Response: Clustering gene expression data allows to identify groups of genes that may act in the same pathway/response and allows us to predict gene function. Our targeted cluster analysis identified 10 clusters characterized by different expression patterns. Three MtKNOX3-like genes placed in Cluster 3 and were characterized by a transient expression peak at 4D. Correlation of centroids identified Cluster 6 as a group of genes with an expression pattern highly anti-correlated to Cluster 3. It has been observed that genes whose expression patterns are strongly anti-correlated can also be functionally related (https://doi.org/10.1093/bioinformatics/btg209), as a gene may be strongly suppressed to allow another to be expressed. This is the case for transcriptional repressors and their target genes that may cluster into separate anti-correlated groups. Since KNOX2 TF are predicted to be transcriptional repressors, we focused on the cluster were most of MtKNOX3-like genes placed (Cluster 3), and on the anti-correlated one (Cluster 6) to predict possible target genes. The other clusters may be more interesting for studying drought response, and were considered in the study of Zhang et al 2014, but our interest was solely to use this excellent transcriptomic experiment to predict possible targets of the TF of our interest that could explain decreased tolerance of RNAi lines. We added this information in the main text to better clarity our bioinformatics approach.

Lines 441-446:      Description may be reduced.

Response: The description was reduced.

Lines 447-449;      The trend the proline biosynthesis genes should have been evaluated along with PDH.

Response: Proline biosynthesis gene expression was analysed in the study of Zhang et al. 2014 and it is not correlated to MtKNOX3-like gene expression. See above.

Lines 456-464: This is important aspects as P5CS roles in proline accumulation has not been addressed in the results, and the proline accumulation cannot be solely attributed to the PDH regulated by KNOX3, which is also not a continuous phenomenon throughout the stress.

Response: We agree that P5CS play a major role in proline accumulation, but our correlation and cluster analysis did not point to any regulatory relationship between MtKNOX3-like TF and P5CS biosynthesis genes.

Lines 472-473:      The RNAi lines accumulate similar amount of proline till 8D and there is not statistical difference. Kindly recheck?

Response: We made this sentence more clear as it is true that difference in proline accumulation is only significant at 12D.

Lines 484-486: Some indicators of oxidative stress should have been analyzed also.

Response: paraquat is an indicator of oxidative stress response and was analysed in our plant material

Material and methods: Comments are given below and also in the PDF file.

Lines 505-506:      Humidity conditions may also be mentioned. Kindly check the units of PFD. the commonly used unit is μmol/m2/s.

Response: We added the information requested (40% relative humidity, 100 μmol/m2/s)

Lines 535-536:      Give conc. in molarity (milli or micro), Indicate Units of enzyme.

Response:  We revised, but no information about the Units are available for the polymerase.

Line 540:                This section may be combined with section 4.1, by appropriately modifying the title. Some repetitive content may be removed.

Response:  We revised accordingly.

Line 544:                Kindly check the units of PFD. the commonly used unit is μmol/m2/s.

Response:  We revised accordingly.

Line 556:                The protocols described under this section may be written in a concise manner and by citing previous publications at appropriate places.

Response:  We revised accordingly. (Adelaide)

Line 596:                What Arabidopsis plants are mentioned here, these are not used in this experiment?

Response:  Sorry, there was a mistake in here. We removed the sentence.

Lines 606-608:      If methods form a previous report is followed kindly cite it.

Response:  We revised accordingly.

Figures: The concerns related to the Figures are indicated in the PDF file and some are listed below.

Figure 1: The significance symbols are not indicated in the Fig 1 C and D panels.

Response:  We revised accordingly.

Figure 2: Figure 2C should be on left side while 2d should be moved to right side and label of 2d missing. Text indicates Fig 2c and 2d, kindly rectify if needed

Response:  We revised accordingly.

Figure 3: Detailed legend description may be moved to the M&M section. Some statements are repetitive as in results as above, It may be removed or description modified.

Response:  We revised accordingly.

Figure 4: The details for GCN analysis are mentioned at several places. This can be minimized and  can be in M&M.

Response:  We revised accordingly.

Figure 5: Kindly check if this data/ figure may need permission from the Authors/Publisher of 2014 paper.

Response:  Transcriptomics data are publicly available and were published. We used the available values to represent the gene expression pattern in the graphs.

Figure 6: IF the levels of PDH are compared in the RNAi lines at T0 and T4 there is not difference although it reduces in watered conditions. The comparisons needs to be done carefully. Why PDH is not getting down-regulated in control plants subjected to drought after 4D time-point. Statistical significance at 4D time-point in control W and Control drought looks very low or appears insignificant (although marked at *). The method used for analysis and actual p values may be indicated)

Response: We agree that comparisons need to be done carefully, and only between plants at the same stage/time-point. Absolute gene expression levels in plants at different developmental stages can be different due to factors other than experimental conditions.

The difference shown are significant and they were measured using a statistical T-test analysis.

Figure 7: The letters are same (a) from T0 to 8D between the two plants, indicating no significant difference? How then the proline accumulation can be visualized in different between the two lines and mediated by KNOX3? Hence role of other three proline biosynthesis genes cannot be excluded, as they are also induced in different stresses including drought.

Response: We agree that the role of the three proline biosynthesis genes cannot be excluded, but the reduced proline accumulation could be an adaptation/homeostasis/feedback response to increased degradation due to lack of downregulation of PDH gene at early stages of drought response. There are no indications that MtKNOX3-like TFs could regulate proline biosynthesis genes (see also other explanations above).

Supplementary Data: Supplementary data files are fine but it should be made sure that the captions or legends should be sufficiently descriptive for easy understanding.

Response: We added captions that are more descriptive.

Round 2

Reviewer 3 Report

No suggestion

Reviewer 4 Report

Manuscript ID: ijms-2506448v2

Title:  A novel role of Medicago truncatula KNAT3/4/5-like class 2 KNOX transcription factors in drought stress tolerance

Authors: Iannelli MA et al.

General comments: The revised manuscript has been considerably improved, and the concerns raised have been appropriately addressed.  Few minor comments may be looked into carefully and addressed, if needed

1)  In the statistical analysis, the number of asterix /stars are indicative of the level of significance. General usage is p-value is less than 0.05 (*), p-value less than 0.01 (**), and p-value is less than 0.001 (***). In the manuscript, Single star (*) is indicates P-value of 0.01 at few places. Kindly check and rectify at Figure 1 legends and also other places, if applicable. At few other places (*) is indicative of P-value of ≤ 0.05. The consistency of symbols to represent the level of significance may be maintained